# Formyl-peptide receptor type 2 activation mitigates heart and lung damage in inflammatory arthritis

Andreas Margraf [1,4], Jianmin Chen [1,4], Marilena Christoforou [1], Pol Claria-Ribas [1], Ayda Henriques Schneider [1], Chiara Cecconello [1], Weifeng Bu [1], Paul R C Imbert [1], Thomas D Wright [1], Stefan Russo [1], Isobel A Blacksell [1], Duco S Koenis [1], Jesmond Dalli, John A Lupisella [2], Nicholas R Wurtz [2], Ricardo A Garcia [2,3], Dianne Cooper [1], Lucy V Norling [1✉] & Mauro Perretti [1✉]

## Abstract

Rheumatoid arthritis (RA) is associated with heart and lung dysfunction. Current therapies fail to attenuate such complications. Here, we identify formyl-peptide receptor type 2 (FPR2) as a therapeutic target to treat heart and lung dysfunction associated with inflammatory arthritis. Arthritic mice on high levels of dietary homocysteine develop cardiac diastolic dysfunction and reduced lung compliance, mirroring two comorbidities in RA. Therapeutic administration of a small molecule FPR2 agonist (BMS986235) to hyper-homocysteine arthritic mice prevented diastolic dysfunction (monitored by echocardiography) and restored lung compliance. These tissue-specific effects were secondary to reduced neutrophil infiltration, modulation of fibroblast activation and phenotype (in the heart) and attenuation of monocyte and macrophage numbers (in the lung). A dual FPR1/2 agonist (compound 43) failed to prevent the reduction in lung compliance of arthritic mice and promoted the accumulation of inflammatory monocytes and pro-fibrotic macrophages in lung parenchyma. This cellular response lies downstream of FPR1-mediated potentiation of CCL2-dependent monocyte chemotaxis and activation. This finding supports the therapeutic development of selective FPR2 agonists to mitigate two impactful comorbidities associated with inflammatory arthritides.

**Keywords** Resolution Pharmacology; Pro-resolving GPCR; HFpEF; Lung Injury; Rheumatoid Arthritis
**Subject Categories** Cardiovascular System; Immunology; Respiratory System

See also: H Shi and BH Annex

## Introduction

Rheumatoid arthritis (RA) is a chronic immune-mediated inflammatory disease associated with joint inflammation and tissue destruction (Weyand and Goronzy, 2021). Characterised by functional impairments, pain, and an approximate 1% penetrance in the general population, RA poses a high financial burden for healthcare systems worldwide (Hsieh et al, 2020). Aside from an impact on the joints, RA is also linked to secondary organ dysfunction and metabolic dysregulation (Roubenoff et al, 1997; Schroecksnadel et al, 2003). Around 30–40% RA patients experience dyspnoea, pleuritis or advancing fibrosis (Norton et al, 2013; Wilsher et al, 2012). Following diagnosis of arthritis-associated interstitial lung disease, a progressive decline in pulmonary function can occur (Zamora-Legoff et al, 2017). Another frequent comorbidity of RA is heart failure with preserved ejection fraction, or HFpEF. Altogether, these co-morbidities are responsible for higher mortality of RA patients (Taylor et al, 2021).

A recent population study confirms a causal association between cardiovascular pathology and RA (Conrad et al, 2022), and highlights the need for experimental models to identify mechanisms and therapeutic opportunities to prevent development of cardiac complications. To address heart failure in arthritis, we have described a transgenic mouse colony that develop arthritis and is characterised by myocardial inflammation, hypertrophy and fibrosis, all processes associated with a HFpEF phenotype accompanied by left ventricular diastolic dysfunction (Chen et al, 2021). Lung compliance is not affected in the transgenic model.

Like other chronic inflammatory diseases, RA is characterised by non-resolving inflammation both in the affected joints and systemically, and this can lead to organ dysfunction, as shown in disease models (Chen et al, 2021; Nathan and Ding, 2010; Rossaint et al, 2021). One way to rectify non-resolving inflammation is to activate tissue resolution processes and attain clinical benefit on joints and other organs in arthritis (Perretti et al, 2017). One prominent pro-resolving factor is formyl-peptide receptor type 2

[1]William Harvey Research Institute, Faculty of Medicine and Dentistry, Queen Mary University of London, London, UK. [2]Department of Cardiovascular and Fibrosis Drug Discovery, Bristol Myers Squibb, Princeton, NJ, USA. [3]Present address: GeneToBe, Ann Arbor, MI, USA. [4]These authors contributed equally: Andreas Margraf, Jianmin Chen. ✉E-mail: l.v.norling@qmul.ac.uk; m.perretti@qmul.ac.uk

(FPR2). Activation of FPR2 in leukocytes and other cells, including stromal cells like fibroblasts and epithelial cells (Perretti and Godson, 2020), regulate inflammatory responses and provides beneficial effects in several disease models, including myocardial infarction and atherosclerosis (Chen et al, 2023; Drechsler et al, 2015; Garcia et al, 2021). Notably, activation of FPR2 and the homologue receptor FPR1 has shown therapeutic benefit against disease symptoms in K/BxN serum transfer-induced arthritis (STIA) (Dufton et al, 2010; Kao et al, 2014). However, FPR1 can also mediate pro-inflammatory responses, including promotion of pulmonary fibrosis (Leslie et al, 2020). Thus, it remains to be established whether selective FPR2 agonism offers superior therapeutic value when compared to non-selective agonists that activate both FPR1 and FPR2.

Since RA patients exhibit metabolic alterations, including elevated plasma levels of homocysteine (Roubenoff et al, 1997; Schroecksnadel et al, 2003), we hypothesised a functional link between hyper-homocysteinemia and lung or heart dysfunction in arthritis. To investigate this, we employed a model of STIA in hyper-homocysteine mice and showed that it recapitulates some of these disease hallmarks. We report that the pathological changes in both lung and heart function can be significantly ameliorated by the therapeutic administration of a selective FPR2 agonist.

# Results

## Characterisation of a novel HH + STIA model

To investigate tissue-specific phenotypic and cellular changes, we established an experimental model where arthritis is associated with both heart and lung dysfunction. Hyper-homocysteinemia (HH) was induced by exposing mice to homocysteine over a 3-week period prior to weekly injections of arthritogenic serum with continuation of homocysteine exposure (Fig. 1A). Induction of HH + STIA led to a time-dependent joint and paw swelling with concomitant reduction of body weight (Fig. 1B–D). HH mice displayed no symptoms of arthritis in the absence of arthritogenic serum administration, although they showed a slight increase in blood pressure (Appendix Fig. S1A,B). Thus, high circulating levels of homocysteine failed to modify arthritis development.

Circulating cell numbers were also quantified across the four groups of animals and showed only minor differences (Appendix Table S1). Appendix Table S1 reports values for the blood clinical chemistry analyses, with no significant differences in creatinine levels, except a modest increase in aspartate aminotransferase.

## HH + STIA is associated with cardiac dysfunction

RA patients are susceptible to HFpEF characterised by cardiac diastolic dysfunction with normal/near-normal ejection fraction (Park and Bathon, 2024). Application of time-course tracking of echocardiography-based cardiac functionality revealed that HH + STIA mice developed diastolic dysfunction as indicated by increased left atrium (LA) area, reduced ratio between the E and A currents (E/A) and moderate increase in deceleration time (Fig. 2A,B). These alterations manifested in neither HH nor STIA groups (Fig. 2A,B). Ejection fraction (EF), an indicator of systolic function, was preserved throughout the entire time course and

confirmed that HH + STIA mice developed cardiac functional abnormalities typical of HFpEF. Quantification of the interventricular septum thickness showed a significant increase in HH + STIA mice, indicating the development of concentric cardiac hypertrophy in these hearts (Fig. 2B). Heart rate was not significantly different amongst groups with values ranging from 400 to 500 bpm for all animals.

Supported by these functional changes, we monitored inflammatory and stromal cells in heart samples. A marked increase in neutrophil infiltration was quantified in HH + STIA mice (Fig. 2C), along with a modest increase in effector T cell numbers (Fig. 2D). We next quantified cardiac fibroblast populations linked with inflammation-associated cardiac dysfunction (Humeres and Frangogiannis, 2019). Both monocytic fibroblasts and structural fibroblasts were increased in HH + STIA cardiac samples, an indication of ongoing fibrotic remodelling (Fig. 2E,F). Congruently, in the same samples we quantified an increase in podoplanin (Pdpn)$^+$Thy1.2$^+$ proinflammatory fibroblasts (Fig. 2G), recently described in the arthritic joint (Croft et al, 2019), along with moderate changes in MEFSK4$^+$Thy1.2$^-$ activated profibrotic myofibroblasts (Fig. 2H). Appendix Fig. S1C reports the gating strategy used for heart cell analyses. Modest variations in monocyte and macrophage numbers, as well as in CD4$^+$ T cells across the four experimental groups, were observed (Appendix Fig. S2A–I).

These experiments indicate that cardiac diastolic dysfunction with preserved ejection fraction develops in HH + STIA, and this functional alteration of the left ventricle is paralleled by changes in specific subsets of immune and fibroblast cells within the myocardium.

## HH + STIA is associated with pulmonary dysfunction

Abnormalities in the respiratory system can be detected in a large proportion of RA patients (Kadura and Raghu, 2021). When we quantified pulmonary function in the experimental groups, only HH + STIA mice displayed substantial pulmonary dysfunction visible as a significant reduction in lung compliance (Fig. 3A). This alteration prompted us to characterise changes occurring within the different pulmonary compartments (Fig. 3B, inset).

Bronchoalveolar lavage white blood cell numbers did not significantly change across the four experimental groups (Appendix Fig. S3A), thus we focussed on lung interstitium. Pulmonary dysfunction is linked to an imbalance in macrophage M1/M2 populations (Rossaint et al, 2021) and we observed an increase in M1/M2 macrophage ratio in HH + STIA mouse lung samples (Fig. 3B), associated with increased CD80 M1-like and lower CD206 M2-like prototypical macrophage markers (Fig. 3C,D). We found a slight elevation of CX$_3$CR1$^+$CD11c$^+$SiglecF$^+$MHCII$^{hi}$ profibrotic macrophages in HH mice, which have been described in the fibrotic lung (Aran et al, 2019), yet values were not significantly different between HH + STIA mice compared to STIA-only or naive animals (Fig. 3E). CD45$^+$ hematopoietic cells, as well as neutrophil numbers in lung interstitium, were not different across the experimental groups (Appendix Fig. S3B,C) and the same was true for other cell types (see Appendix Fig. S3D,E). The flow cytometry gating strategy of lung samples is shown in Appendix Fig. S4.

Having characterised HH + STIA mice as an experimental setting where joint disease is accompanied by clinically relevant

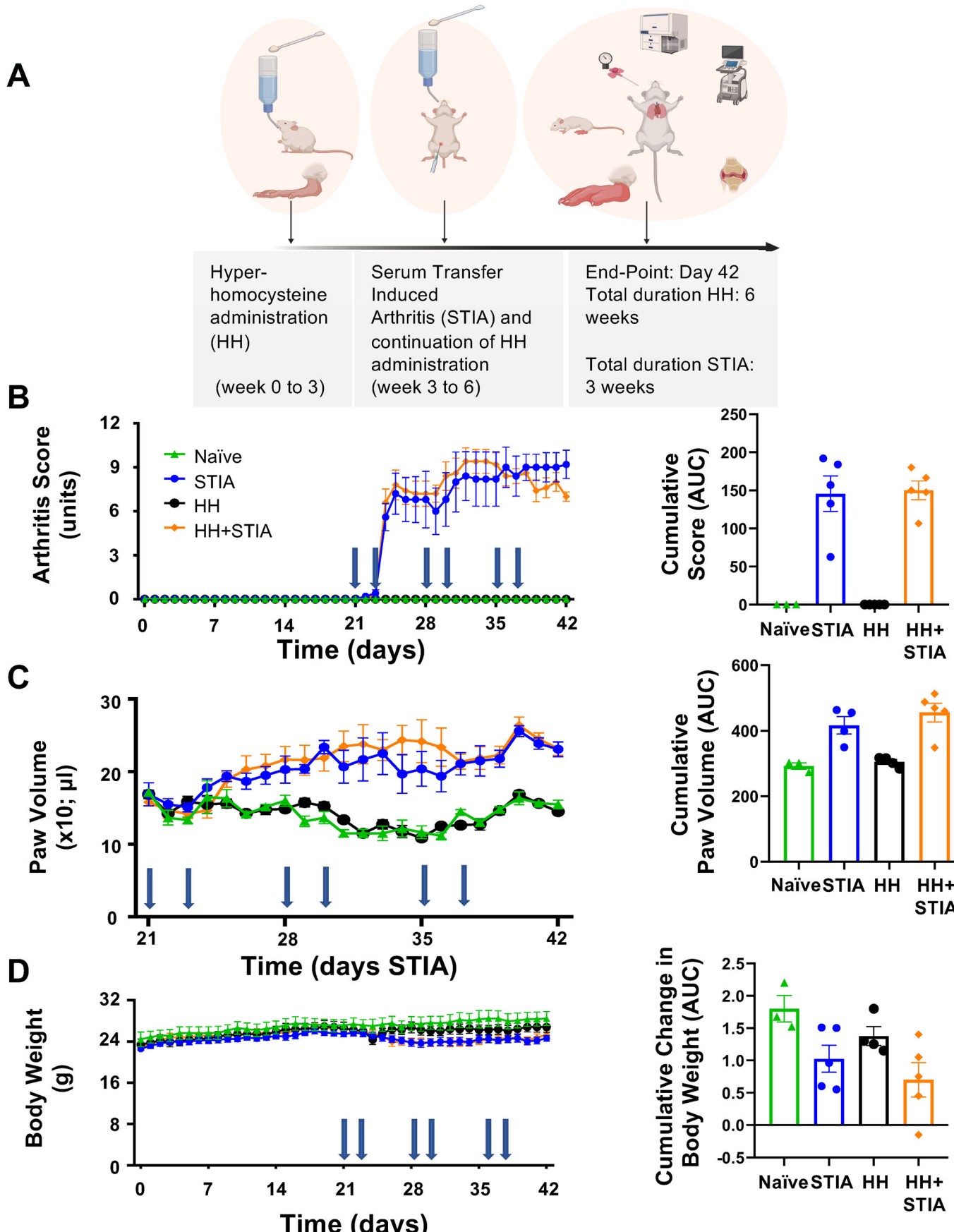

◄ **Figure 1. A model of hyper-homocysteinemia plus serum transfer induced arthritis (HH + STIA).**

(A) Mice were treated with ad libitum access to homocysteine-supplemented drinking water for a total duration of 6 weeks. After the initial 3 weeks (hyper-homocysteine [HH] induction period), arthritogenic serum transfer-induced arthritis (STIA) was initiated with injections at day 0 and day 2 of each week for a total of 3 weeks. Mice were culled after a total time interval of 6 weeks. (B) STIA is associated with a steady increase in arthritic scoring, while treatment with HH itself does not evoke any symptoms. (C, D) HH + STIA is associated with paw oedema (C) and a reduction in body weight gain (D), with no difference from STIA-only animals. Data are mean ± SEM of $n = 3$ (naive), $n = 4$ (HH) and $n = 5$ (STIA and HH + STIA) mice. Multiple measurements: two-way ANOVA with Tukeys multiple comparisons test. Cumulative statistics: one-way ANOVA with Tukey's multiple comparisons test, showing STIA vs. HH + STIA. Source data are available online for this figure.

functional alterations in the heart and lung, we next explored potential therapeutic options.

## Agonists at formyl-peptide receptors ameliorate cardiac dysfunction in HH + STIA

To identify putative targets of therapeutic relevance in human RA patients and based on recent advances in the field, with combinatorial approaches for comparison of datasets across experiments (Platzer et al, 2019), we utilised a publicly available database of bulk RNAseq data from isolated peripheral blood mononuclear cells (PBMCs) of healthy and RA donors (Shen et al, 2022). Side-by-side comparison of differentially expressed genes revealed up-regulation of *S100A9* and *TNFAIP6* (Fig. 4A), both known proinflammatory signalling components. A significant increase in *FPR2* mRNA expression in RA PBMCs was identified, suggesting potential utility for FPR2 agonists. Further support for this target derived from the presence of this receptor on CD45$^+$ blood cells of HH + STIA mice (Fig. 4B, vehicle group). Even more relevant are the observations that (i) *Fpr2* null mice develop early onset diastolic dysfunction, which is associated with reduced survival upon ageing due to a prolonged failure of resolution responses (Tourki et al, 2020); (ii) human coronary artery endothelial and smooth muscle cells from heart samples express FPR2 as shown by immunohistochemistry (Bouhadoun et al, 2023). Moreover, we monitored the expression of Fpr2 on cardiac cells from HH + STIA mice: Figure EV1 shows high cell surface expression of FPR2 on myeloid cells, more precisely cardiac macrophages and neutrophils, whereas expression was lower on endothelial and fibroblast populations in the heart.

Altogether these data prompted us to test the potential protective effect of FPR2 activation in HH + STIA mice. In doing so, we compared the pharmacological profile of two small molecules: the selective FPR2 agonist BMS986235 (abbreviated to BMS235) and the dual FPR1/FPR2 agonist compound 43 (C43). Compounds were administered along a therapeutic protocol, starting daily oral treatment one week after the induction of STIA (Appendix Fig. S5A). At the end of the treatment, echocardiography analyses revealed significant improvement of both LA area and E/A ratio, compared to vehicle, following treatment with either compound (Fig. 4C–E). Deceleration time was not reduced in the treatment groups (Fig. 4F). Ejection fraction remained unmodified by disease and treatment (Fig. 4G) and, similarly, no changes in left ventricular internal diameter and interventricular septum emerged across the experimental groups (Appendix Fig. S5B).

Multiplex analyses of circulating factors and biomarkers in plasma samples of the experimental groups showed no significant changes following treatment with either BMS235 or C43 (Table

EV1). Thus, at least at the time point under consideration, these small molecules fail to regulate circulating levels of inflammatory mediators. Conversely, tissue-specific functional alterations may derive from direct effects on cells within the relevant tissue.

Since cardiac neutrophil and fibroblast populations were modified in the hearts of HH + STIA mice when compared to HH or STIA groups, we focused on these two cell types for the next series of analyses. Both selective FPR2 agonist and FPR1/FPR2 dual agonist reduced the extent of infiltrated neutrophils in the hearts by approximately 40-50% (Fig. 5A). Analyses of cardiac fibroblast subtypes indicated that BMS235 reduced numbers of monocytic fibroblasts (cTnT$^-$LiveCD31$^-$Lin$^-$CD45$^+$CD34$^+$Thy1.2$^+$ cells; Fig. 5B). C43 also impacted fibroblast subsets with a significant effect on CD45$^-$CD34$^-$Thy1.2$^+$ structural fibroblasts and pathogenic MEFSK4$^+$Thy1.2$^-$ profibrotic subtypes (Fig. 5C,D). Both treatments reduced Pdpn$^+$Thy1.2$^+$ proinflammatory fibroblasts (Fig. 5E). At this time point, the effect of the pharmacological treatments on cardiac monocytes, macrophages and T cells was modest (Appendix Fig. S2J–S). In all cases, these effects were not secondary to overt changes in blood pressure by either compound (Appendix Fig. S1A,B).

Since repeated receptor activation may lead to long-lasting receptor downregulation, we performed flow cytometric analysis of FPR2 surface levels in circulating CD45$^+$ cells and observed no change or rather a modest increase in cell-associated FPR2 following 2-week treatment with both agonists (Fig. 4B; BMS and C43 groups). Of the same vein, treatment with BMS235 did not modify FPR2 expression of cardiac cells either (Fig. EV1).

Finally, BMS235 was also tested in a different model of arthritis which is characterised by diastolic dysfunction, the K/BxN F1 mouse colony which spontaneously develops this comorbidity (Chen et al, 2021). Daily treatment of mice over a 4-week period with BMS235 (3 mg/kg p.o.) reduced the parameters of diastolic dysfunction (Fig. EV2). This prophylactic cardio-protection occurred without significantly affecting severity of arthritis (Fig. EV2G).

In conclusion, both BMS235 and C43 improved cardiac function, which is compromised in HH + STIA. The cardio-protective effect of BMS235 was also validated in K/BxN F1 mice. In HH + STIA, this effect was associated with modulation of fibroblast phenotypes and reduction in neutrophil numbers. Next, we tested whether these pharmacological treatments could impact lung function.

## Selective FPR2, but not dual FPR1/2, activation alleviates pulmonary dysfunction

Treatment of HH + STIA animals with BMS235 improved lung compliance, restoring it to levels of naive animals (dashed line),

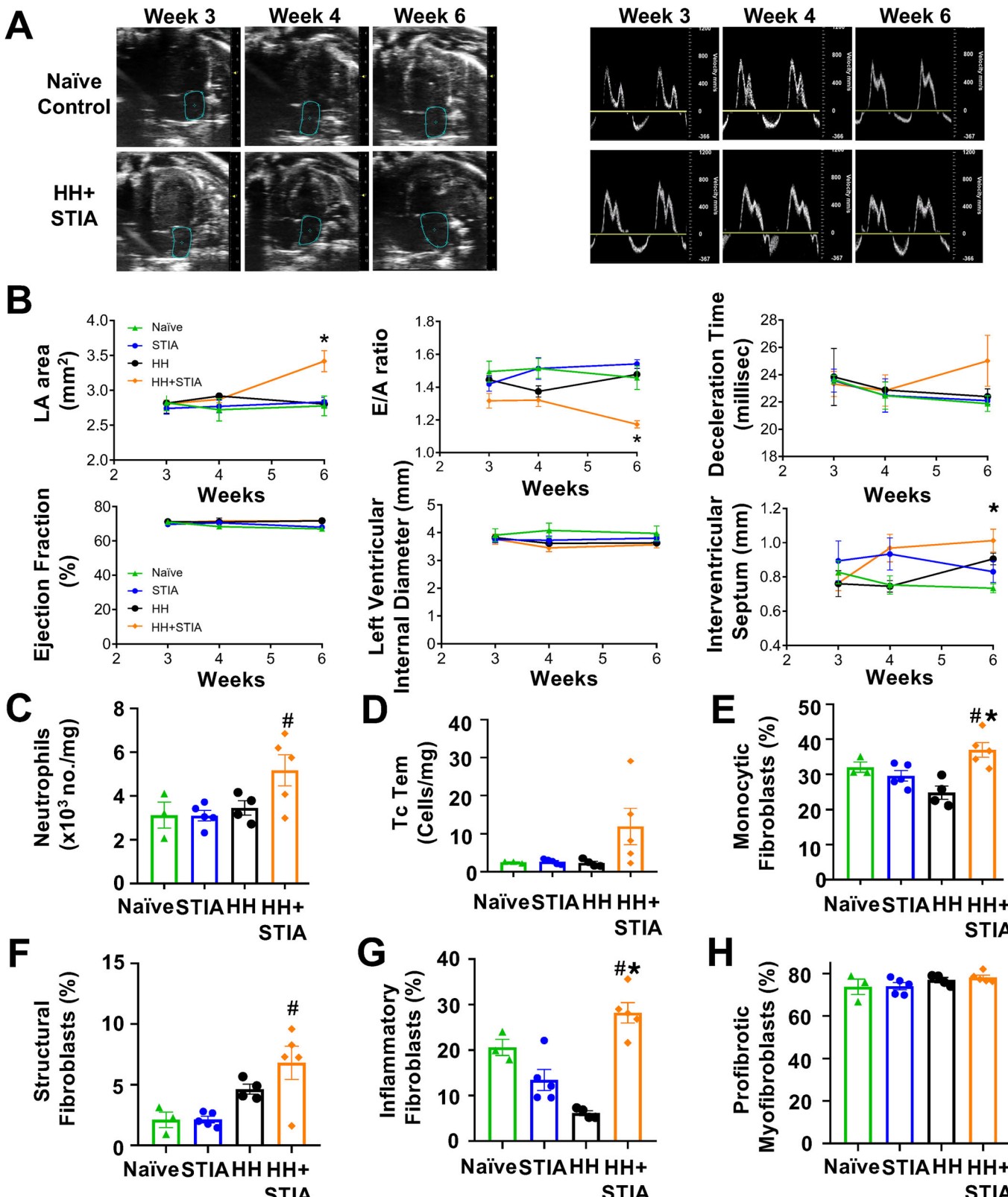

**Figure 2.** Hyper-homocysteinemia plus serum transfer-induced arthritis (HH + STIA) is associated with cardiac dysfunction.

Application of HH + STIA treatment (see Fig. 1) leads to a progressive impairment in cardiac diastolic functionality. (**A**) Left-hand images: representative B-mode four-chamber echocardiograms and increased left atrial (LA) area in arthritic mice. Right-hand images: representative mitral flow patterns from pulsed-wave colour Doppler echocardiography. (**B**) Quantification of six cardiac parameters across the four experimental groups; HH + STIA display increased left atrial (light blue circle in (A)) area (*P = 0.0060); reduced E/A ratio (*P = 0.00000025); interventricular septum thickness shows significant interaction between time and treatment (*P = 0.0325); no change in ejection fraction and left ventricle internal diameter, with a trend for increased deceleration time. (**C**) neutrophils: CD45+7/4+Ly6G +; adjusted P value = 0.0380. (**D**) T effector memory cells: CD4+CD62L-CD44+. (**E**) monocytic fibroblasts: cTnT-CD31-Lin-CD45+CD34+Thy1.2+ cells; adjusted P value = 0.0395. (**F**) structural fibroblasts: cTnT-CD31-Lin-CD45-CD34-Thy1.2 +; adjusted P value = 0.0066. (**G**) inflammatory fibroblasts: cTnT-CD31-Lin-Pdpn+Thy1.2 +; adjusted P value = 0.0005. (**H**) activated profibrotic myofibroblasts: cTnT-CD31-Lin-CD45-CD34-MEFSK4+Thy1.2-. Data are mean ± SEM of n = 3 (naive), n = 4 (HH) and n = 5 (STIA and HH + STIA) mice. (**B**) * Denotes significant interaction between treatment and time (P ≤ 0.05). (**C–H**) #P < 0.05 vs. STIA group. *P < 0.05 vs. HH group. (**B**) Two-way ANOVA with Bonferroni's multiple comparisons test. (**C–H**) One-way ANOVA with Tukey's multiple comparisons test. Source data are available online for this figure.

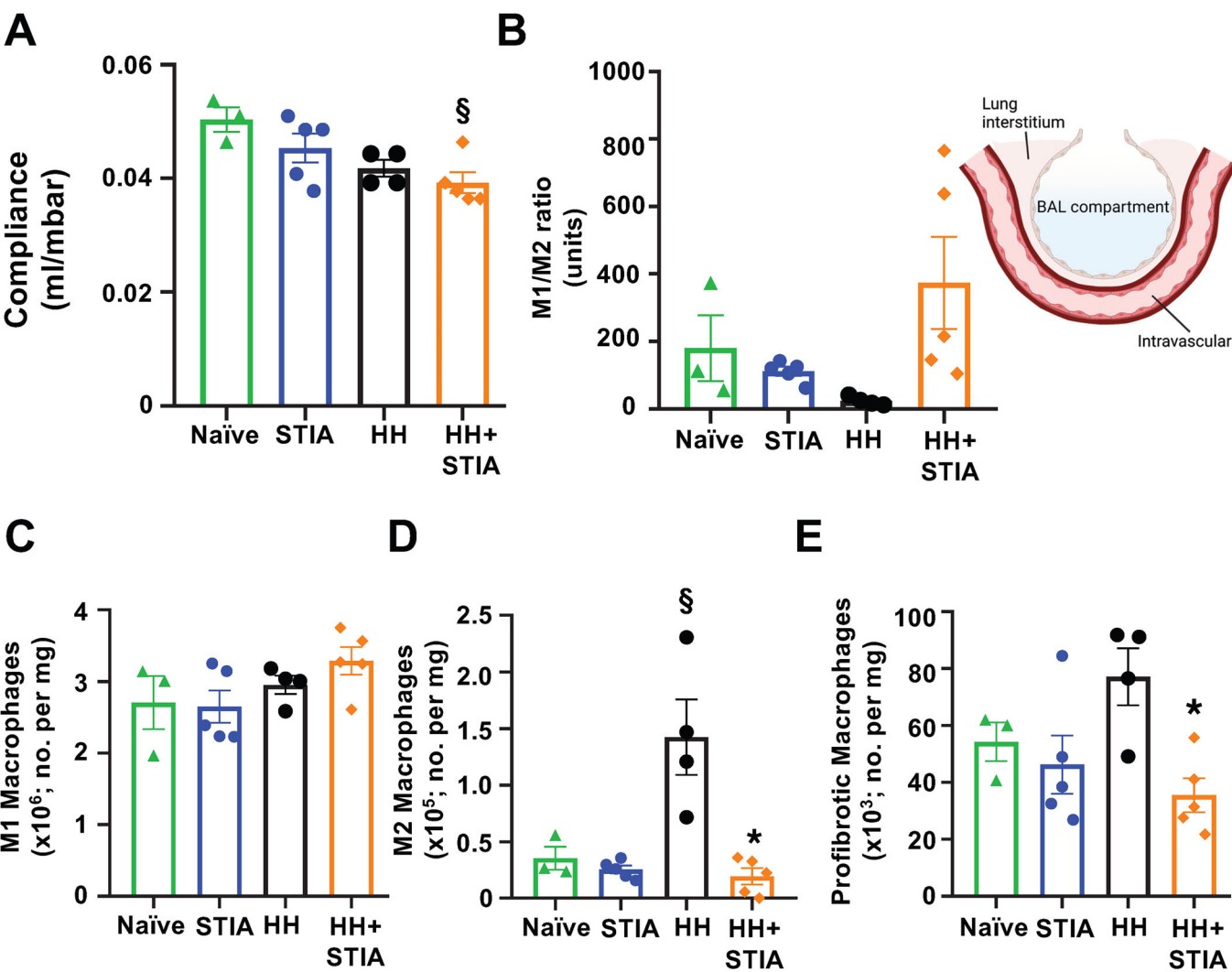

**Figure 3.** Hyper-homocysteinemia plus serum transfer-induced arthritis (HH + STIA) is associated with pulmonary dysfunction.

Application of HH + STIA treatment (see Fig. 1) leads to a progressive decline in pulmonary function and immune cell numbers. (**A**) Lung compliance as measured at day 42; adjusted P value = 0.0198. (**B–E**) Interstitial immune cell profiling of the lungs collected on day 42. Macrophage subtypes were identified as follows. M1-like macrophages: CD45+CX3CR1+F4/80+CD80+CD206low; M2-like macrophages, CD45+CX3CR1+F4/80+CD80-CD206high; profibrotic macrophages, CX3CR1+CD11c+SiglecF+MHCIIhigh. Analyses revealed a slightly elevated M1/M2 ratio (**B**) with a significant increase in M1 macrophages (**C**) and differentially modulated M2 profiles (§, adjusted P value = 0.0010; *, adjusted P value = 0.0006) as well as profibrotic macrophages in the HH group (*, adjusted P value = 0.0207) (**D, E**). Data are mean ± SEM of n = 3 (naive), n = 4 (HH) and n = 5 (STIA and HH + STIA) mice. §P < 0.05 vs. naive mice; *P < 0.05 vs. HH group. One-way ANOVA with Tukey's multiple comparisons test. Source data are available online for this figure.

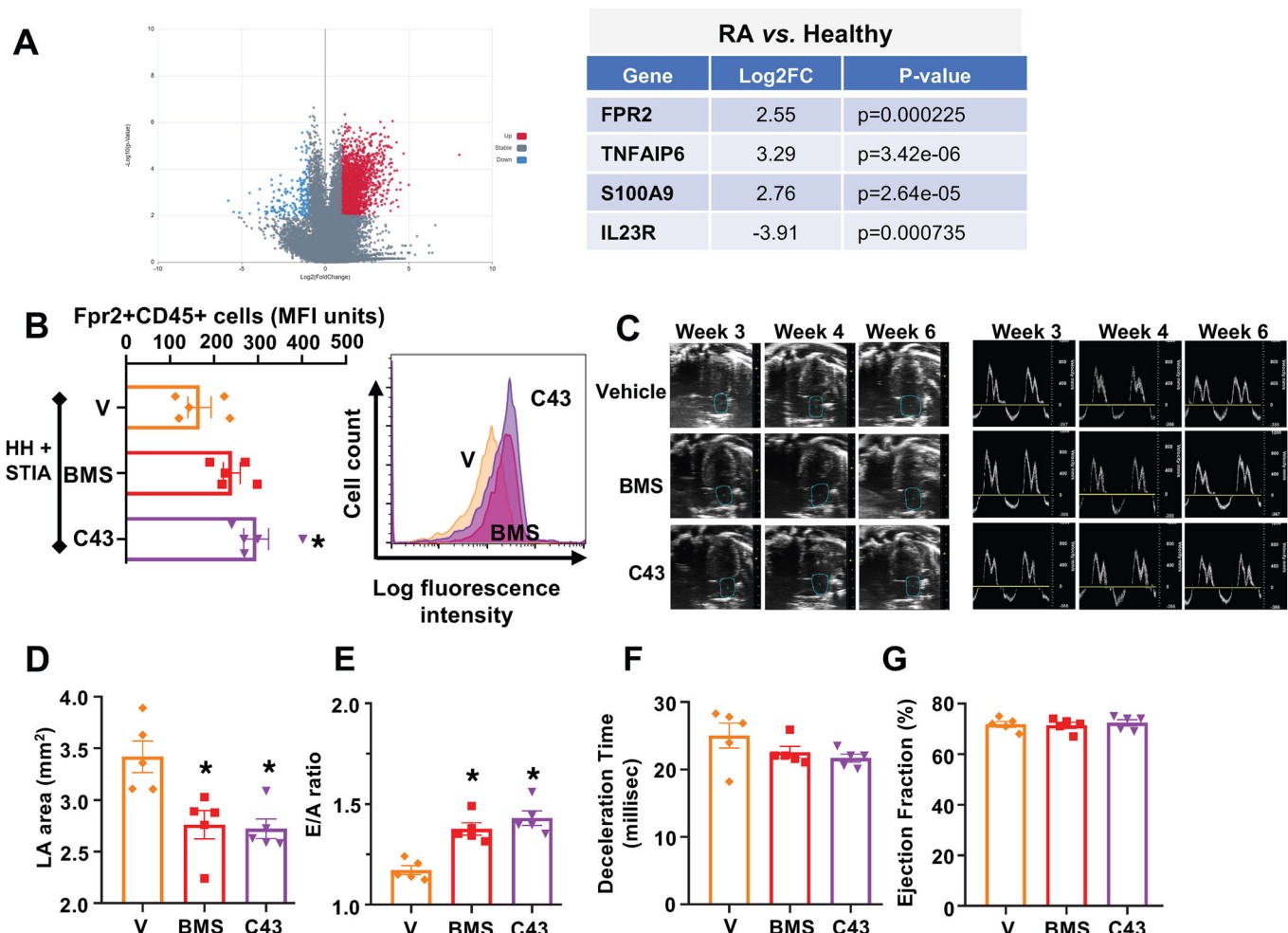

**Figure 4. Target screening and functional impact of agonists at formyl-peptide receptors on cardiac dysfunction in HH + STIA.**

(A) A publicly available database for RNAseq datasets of autoimmune maladies was interrogated for bulk RNAseq data on PBMCs isolated from healthy vs. rheumatoid arthritis (RA) patients. Volcano plot (left panel) shows significant changes in gene expression patterns. Selected disease-relevant genes are presented in the table, and display increased expression levels of FPR2, TNFAIP6, S100A9 and a reduction in IL23R. (DEG method: Wilcoxon; P value cutoff: 0.01; RA vs. Healthy; Sample count: n = 12 RA and n = 64 healthy datasets; see: (Shen et al, 2022)). (B) Fpr2 expression in CD45+ blood cells of HH + STIA mice (see Fig. 1), treated daily for the last two weeks with vehicle, BMS235 (3 mg/kg p.o.) or C43 (10 mg/kg p.o.). Left: quantitative data presented as median fluorescence intensity (MFI) units; right: representative histograms for Fpr2 distribution in CD45+ cells (*, adjusted P value = 0.0085). (C) Left-hand images: representative B-mode four-chamber echocardiograms and left atrial (light blue circles) area in HH + STIA mice. Right-hand images: representative mitral flow patterns from pulsed-wave colour Doppler echocardiography. (D–G) Quantification of four cardiac parameters across the experimental groups; (D) left atrial (LA) area (*, adjusted p-value for BMS vs. V = 0.0099; *, adjusted P value for C43 vs. V = 0.0069); (E) E/A ratio (*, adjusted P value for BMS vs. V = 0.0011; *, adjusted P value for C43 vs. V = 0.0001); (F) deceleration time; (G) ejection fraction. Data are mean ± SEM of n = 5 mice per group. *P < 0.05 vs. vehicle group. One-way ANOVA with Tukey's multiple comparisons test. Source data are available online for this figure.

whereas the dual FPR1/2 agonist was ineffective (Fig. 6A). Characterisation of interstitial macrophage subtypes and monocytic cells in HH + STIA lungs revealed an interesting scenario. Both the selective FPR2 agonist and the non-selective FPR1/2 dual agonist mildly decreased the M1/M2 ratio (Fig. 6B). This change was due to a significant reduction in M1 macrophage numbers (Fig. 6C) rather than an elevation of M2 macrophages (Fig. 6D). To understand why macrophage regulation did not translate into beneficial compliance changes following FPR1/2 agonist treatment, we expanded our analyses.

No difference emerged for interstitial neutrophils, CD4+ or CD3+ T cells (Appendix Fig. S3F–H). However, the lungs of mice treated with C43 presented elevated numbers of Ly6Chi classical monocytes, double that quantified in vehicle- and BMS235-treated HH + STIA groups (Fig. 6E). C43 reduced non-classical monocytic cells in the lung interstitium (Fig. 6F) and resulted in an approximately threefold increase in CCR2 expression on F4/80+ macrophages (Fig. 6G upper and lower panel). Finally, we monitored the profibrotic interstitial macrophage population (Aran et al, 2019): CX3CR1+CD11c+SiglecF+MHCIIhi cells were increased selectively in the C43 group of mice, with no difference between BMS235 and the vehicle group (Fig. 6H).

Platelet-leukocyte aggregates are typical of active inflammation, and this intercellular interaction can propel further pro-inflammatory

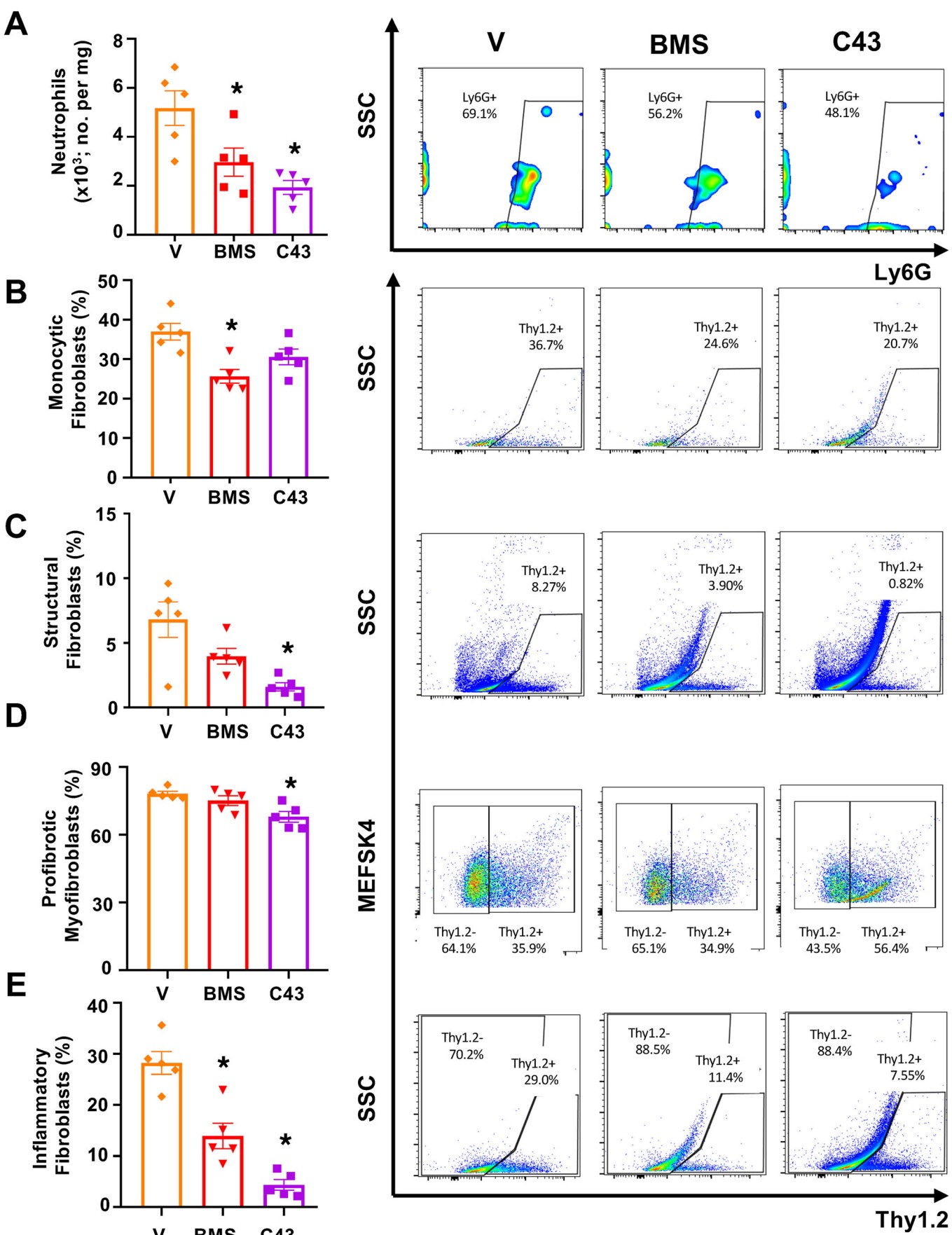

**Figure 5.** Modulation of cellular profiles by FPR-agonism in HH + STIA-mediated cardiac dysfunction.

HH + STIA was induced as in Fig. 1. From week 4, mice were treated with either vehicle or BMS235 (3 mg/kg per os) or C43 (10 mg/kg per os) daily. **(A)** neutrophils: CD45+7/4+Ly6G+ (*, adjusted P value for BMS vs. V = 0.0381; adjusted p-value for C43 vs. V = 0.0036). **(B)** Monocytic fibroblasts; Thy1.2+cTnT-CD31-Lin-CD45 + CD34+ cells (*, adjusted P value = 0.0027). **(C)** Structural fibroblasts: Thy1.2+cTnT-CD31-Lin-CD45-CD34- (*, adjusted P value = 0.0140). **(D)** Activated profibrotic myofibroblasts: MEFSK4+Thy1.2-cTnT-CD31-Lin-CD45-CD34- (*, adjusted P value = 0.0070). **(E)** Inflammatory fibroblasts: Thy1.2+cTnT-CD31-Lin-Pdpn+ (*, adjusted P value for BMS vs. V = 0.0008; adjusted P value for C43 vs. V < 0.0001). Right hand side: representative flow cytometry plots for each experimental group. Data are mean ± SEM of $n = 5$ mice per group. *$P < 0.05$ vs. vehicle group. One-way ANOVA with Tukey's multiple comparisons test. For **(C)**: Kruskal–Wallis ANOVA followed by Dunn's multiple comparisons test since prior testing with Bartlett's test (as performed for all analyses) showed a P value of 0.0325. Source data are available online for this figure.

mechanisms (Margraf and Zarbock, 2019). Therefore, we evaluated whether the immuno-regulatory effects of FPR-based therapeutics in the lung could impact on this marker of tissue inflammation. Platelet-leukocyte aggregates were detected in HH + STIA mice with a lower proportion of these aggregates in BMS235-treated mice compared with C43-treated mice (Fig. 6I). No significant difference in BAL-infiltrating white blood cells was notable across all groups (Appendix Fig. S3I).

Prompted by C43-induced changes in pro-fibrotic cells, we next performed molecular analyses of specific pro-fibrotic factors. Nestin, a type VI intermediate filament protein, is a marker of pulmonary dysfunction and fibrosis as it facilitates TGF-β receptor recycling (Wang et al, 2022). Quantification of nestin mRNA in HH + STIA mouse lung samples showed an approximately 40% increase compared to naive, HH or STIA lung samples, reinforcing its diagnostic value ($P < 0.01$ vs. HH and $P < 0.05$ vs. STIA; $n \geq 3$). However, lungs from mice treated with BMS235 presented ~50% reduction in nestin mRNA compared to vehicle-treated mice, an effect not replicated in C43-treated mouse lungs (Fig. 6J). Figure 6K illustrates the heatmap for selected genes involved in fibrosis (e.g. *Spp1, Col1a1, Tgf-b, Nestin* and *Fra2*) or inflammation (e.g. *Fpr2* and *Il6*) as quantified by qPCR from lung extracts. Of interest, genes that were activated in HH + STIA as compared to STIA alone, e.g. *Il6* and *Col1a1*, were reduced in both BMS235 and C43 groups.

## Imaging of hearts and lungs from HH + STIA mice treated with BMS235

Both flow cytometry and RNA data demonstrate increases in fibroblast or fibrosis-related cell populations and genes. To assess organ fibrosis, we applied multiphoton microscopy of heart and lung slices. Using second harmonic generation (SHG) imaging, we assessed changes in structure and composition of collagen fibres within these organs. Herein, increased collagen content could be detected in HH + STIA mice. Upon administration of BMS235, these changes were markedly reduced with more prominent effects visible in hearts compared to lungs (Fig. 7A–F). A similar trend for cardiac fibroblasts was observed by immunofluorescence spinning disk microscopy (Fig. 7G,I), with a higher proportion of vimentin-positive fibroblasts in HH + STIA over naive heart sections, and a mild attenuation following treatment with BMS235. In these cardiac sections, galectin-3-positive macrophages were observed across the entire interstitium (Fig. 7H,I). In line with the SHG data, less conclusive modulations were observed in lung tissue sections, for both neutrophil and macrophage numbers (Fig. 7J–L). Appendix Figures S6 and S7 present individual channels and composite images for each experimental group.

## Dual agonism of FPR1/2 potentiates human monocyte chemotaxis to CCL2

As we observed changes in CCR2$^+$ expression in lung homogenates from C43-treated mice together with monocyte/macrophage populations, we studied human monocyte chemotaxis to its ligand CCL2. This chemokine is central to monocyte recruitment to the pulmonary tissue and causative for the ensuing lung inflammation (Maus et al, 2002; Rose et al, 2003). Following initial concentration-response experiments of monocyte chemotaxis, the 30 nM concentration of CCL2 was selected (Fig. 8A), to enable positive and negative modulation of the cellular response. Pre-treatment of human monocytes with C43 increased both the rate and magnitude of monocyte chemotaxis to CCL2, an effect not produced with BMS235 (Fig. 8B). Based on these data, we selected C43 to identify molecular mechanisms responsible for this potentiation of the monocyte response.

Firstly, we noted the potentiating effect of C43 was specific for CCL2, as this compound inhibited rather than increased monocyte chemotaxis to C5a (Fig. EV3). Potentiation of the CCL2 response was not secondary to alterations in CCR2 recycling, as expression of this receptor on monocyte plasma membrane was reduced by CCL2 application irrespective of C43 pre-incubation (Fig. 8C). However, C43 altered receptor function. Quantification of β2-integrin neo-epitope indicated additive and/or synergistic effects of C43 on the CCL2 response, with optimal responses at 10 nM CCL2 and 30 nM C43 (Fig. 8D for exemplary experiment; increases in three different donors in Fig. EV3B). Such an additive outcome post-C43 + CCL2 application to monocytes also emerged when we assessed receptor signalling. We screened the phosphorylation of relevant targets and observed a CCL2-dependent phosphorylation of the MAPK pathway with an additive response for the p44/42-p90RSK-S6 pathway, significant for p44/42 phosphorylation (Fig. 8E; Appendix Fig. S8 for the original blots produced with three different donors).

Finally, we tested the direct action of C43 on human monocyte chemotaxis and observed a significant response at 100 nM; such an effect was reliant on both FPR1 and FPR2 (Fig. EV3C shows exemplary data). Out of five distinct cell donors, the FPR1 antagonist cyclosporin H reduced C43 chemotaxis by 67.4 ± 19%, and the FPR2 antagonist WRW4 by 67.3 ± 17% ($P < 0.05$ in both cases). In two distinct cell preparations, the two antagonists together reduced the human monocyte chemotactic response to C43 by 90.2 ± 1%.

A separate set of experiments addressed the potential direct effect of BMS235 on macrophages, cardiac or synovial fibroblasts and their crosstalk. These experiments were conducted with human cells, to increase the translational value of these data. When we cultured human macrophages with either fibroblast type, BMS235

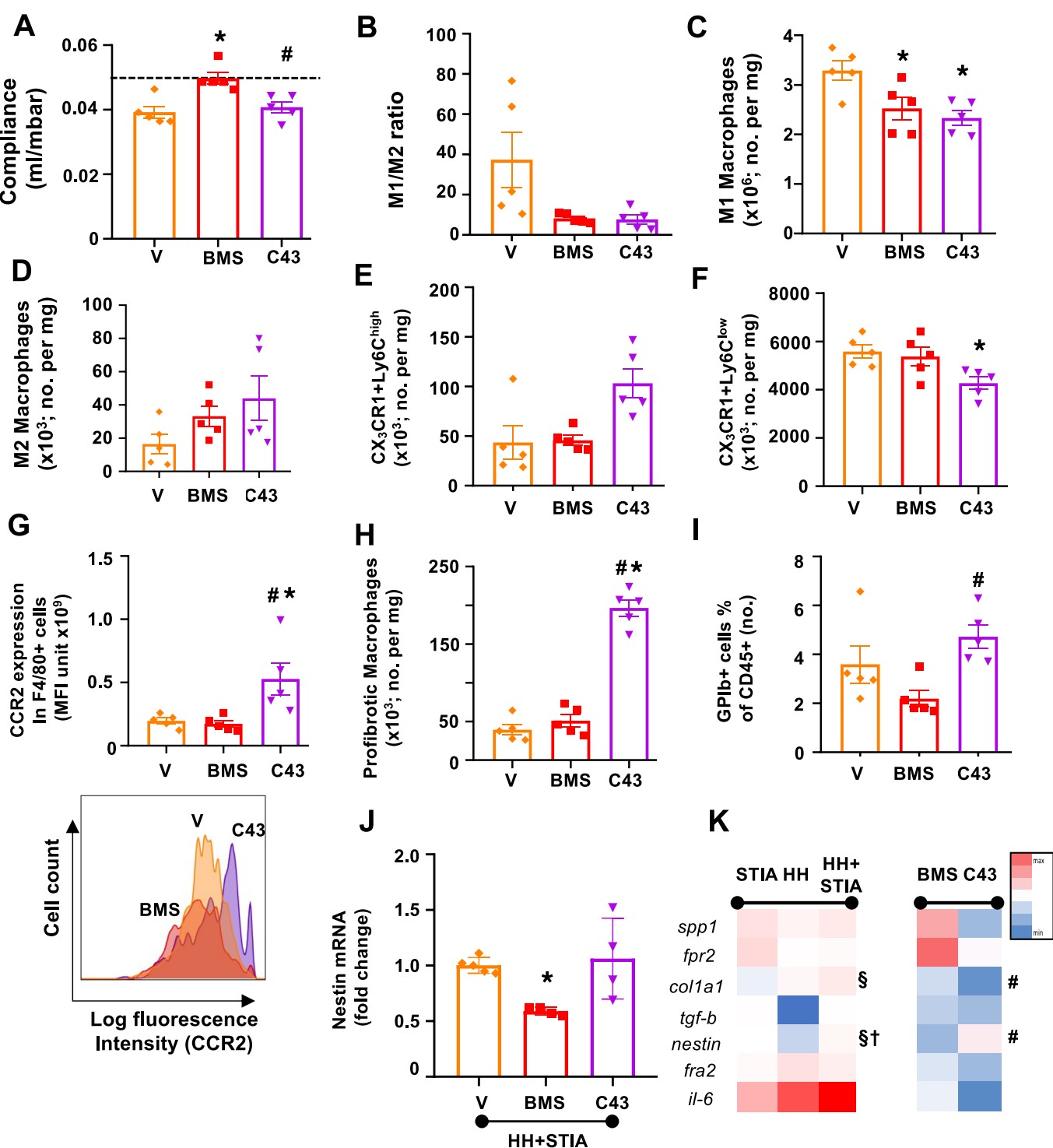

incubation with the macrophage impacted on markers of fibroblast activation in a selective manner. Figure EV4B,C illustrate the indirect inhibition of synoviocytes by BMS235 (shown as reduction in the proportion of VCAM1+ cells; Fig. EV4B) as well as cardiac fibroblasts (shown as a reduction in CCL2 mRNA expression; Fig. EV4C). Altogether these data indicate that BMS235 can interfere with cell-to-cell interactions that may occur in diseased tissues. Further experiments are required to substantiate this

finding and to expand its value in relation to future translational development.

## Selective FPR2 but not dual FPR1/2 agonism reduces joint disease

Finally, we monitored joint disease to evaluate whether FPR agonists reduced arthritis inflammation in both joint and distant

**Figure 6. Functional impact of agonists at formyl-peptide receptors on pulmonary dysfunction in HH + STIA.**

HH + STIA was induced as in Fig. 1. From week 4, mice were treated with either vehicle or BMS235 (3 mg/kg per os) or C43 (10 mg/kg per os) daily and analyses were conducted at week 6 (day 42). (A) Lung compliance (*, adjusted $P$ value = 0.0035; #, adjusted $P$ value = 0.0103). (B-I) Cellular characterisation from lungs; (B–D) M1 macrophages: CD45+CX3CR1+F4/80+CD80+CD206low (*, adjusted $P$ value for BMS vs. V = 0.0396; adjusted $P$ value for C43 vs. V = 0.0112); M2 macrophages: CD45+CX3CR1+F4/80+CD80-CD206high. (E) Classical monocytes: CX3CR1+Ly6Chigh. (F) Non-classical monocytes: CX3CR1+Ly6Clow (*, adjusted $P$ value = 0.0296). (G) CCR2 expression on F4/80+ cells (*, adjusted $P$ value = 0.0249; #, adjusted $P$ value = 0.0165; lower part: representative histogram with overlapping profiles for one sample from each group. (H) Profibrotic macrophage population: CX3CR1+CD11c+SiglecF+MHCIIhigh (*, adjusted $P$ value = 0.00000006; #, adjusted $P$ value = 0.00000015). (I) Platelet-leukocyte aggregates as proportion of the CD45+ population (#, adjusted $P$ value = 0.0188). (J) Nestin mRNA as quantified by qPCR (*, adjusted $P$ value = 0.0382). (K) Heatmap of selected gene expression values quantified by qPCR and normalised to naive (left panel) or vehicle (right panel), respectively. Data are mean ± SEM of $n$ = 3–5 mice per group. *$P$ < 0.05 vs. vehicle group; #$P$ < 0.05 vs. BMS group; §$P$ < 0.05 vs. STIA; †$P$ < 0.05 vs. HH. One-way ANOVA with Tukey's multiple comparisons test. For (B, J): Kruskal–Wallis ANOVA followed by Dunn's multiple comparisons test. Source data are available online for this figure.

organs. Treatment with BMS235 reduced arthritic score (Fig. EV5A,B) and paw oedema (Fig. EV5C), with body weight increasing in line with the vehicle group (Fig. EV5D). On all these macroscopic parameters, treatment with C43 was either ineffective or resulted in a slight worsening (e.g. reduced gain in body weight; Fig. EV5D). Since we observed therapeutic effects solely in BMS235-treated animals, joints from these animals were further analysed for cell components. Treatment with BMS235 did not modify the numbers of CD45$^+$ cells, classical or non-classical monocyte populations but yielded a significant reduction in joint-infiltrating neutrophils compared to vehicle treatment (Fig. EV5E). The number of aggressive joint fibroblasts, a phenotype recently described in human RA synovia (Croft et al, 2019) and characterised as CD45$^-$CD31$^-$Pdpn$^+$Thy1.2$^+$Fap$^+$ cells, was reduced by ~50% following treatment with the selective FPR2 agonist (Fig. EV5E). The gating strategies applied for the flow cytometry analyses of paw samples are shown in Appendix Fig. S9.

## Discussion

In this study, we provide evidence for organ-specific immune mechanisms as drivers of organ dysfunction in settings of inflammatory arthritis, a conclusion reached through side-by-side comparisons of joints, hearts, and lungs, and supported by the use of pharmacological tools that activate formyl-peptide receptors. With the model of HH + STIA, we recapitulate specific features of RA patients—as presented in the introduction—and provide proof of concept that selective agonists at FPR2 can offer a therapeutic opportunity, not only to control joint disease but also control the onset of secondary organ injuries which, in patients, represent a major comorbidity and can be lethal.

In addition to joint pain and reduced joint motion, RA patients present extra-articular disease manifestations in the lungs and heart (Figus et al, 2021). These extra-articular manifestations are often associated with increased homocysteine levels (Gazar et al, 2020), but the underlying mechanisms, immunological changes and therapeutic strategies against secondary organ alterations in RA are so far insufficiently appreciated (Kadura and Raghu, 2021). Thus, in this study, we set out to (i) determine organ-specific pathophysiological cell profiles and functional consequences in HH + STIA and (ii) assess the therapeutic and immunomodulatory effects of an FPR2 agonist. Application of the HH + STIA model provided important outcomes: (i) unaltered arthritis by HH; (ii)

presence of heart and lung functional alterations; (iii) distinct changes in immune and stromal cell populations specific for the heart or the lung; (iv) lack of changes in systemic markers of inflammation. Thus, this model could be used to fill the experimental gap necessary to address the unmet clinical need discussed above (Conrad et al, 2022).

RA-associated cardiomyopathy is not managed by current therapies. As an example, anti-TNF therapy may even adversely affect incidence of heart failure and death (Chung et al, 2003). In the same vein, therapeutic modalities for interstitial lung disease in RA are of limited efficacy and complicated by pulmonary toxicity (Kadura and Raghu, 2021). Here, we tested a novel potential therapy that targets FPR2: this pro-resolving receptor is highly abundant in myeloid cells (Perretti and Godson, 2020) and is also expressed in other cell types relevant to the organs investigated here, namely fibroblasts, endothelial and epithelial cells, as shown with murine cells here and reported in the literature (Chen et al, 2023; Drechsler et al, 2015; Garcia et al, 2021). Of note, human heart samples also express FPR2 (Bouhadoun et al, 2023). In terms of activity, the FPR2-specific agonist BMS235 and FPR1/FPR2 dual agonist C43 afforded only partial overlapping organ protection and modulation of cellular responses. The diastolic dysfunction associated with inflammatory arthritis was controlled to a similar extent by BMS235 and C43. This shared efficacy on cardiac functionality was reflected in a similar modulation of immune and fibroblast cell subsets in the heart, with significant effects on neutrophil infiltration and cardiac fibroblasts, especially the Pdpn$^+$Thy1$^+$ proinflammatory fibroblast type, recently described to be particularly aggressive in the synovia (Croft et al, 2019). The protection against diastolic dysfunction afforded by BMS235 was also observed in KBxN F1 mice, which are known to develop arthritis spontaneously 4 to 5 weeks before evidence of heart alterations (Chen et al, 2021). Our findings here, coupled with the studies investigating FPR small agonists in myocardial infarction (Garcia et al, 2021; Qin et al, 2017), identify cardiac syndromes as ideal disease targets for this family of pharmacologically active compounds.

At variance from the cardiac analyses, the selective FPR2 agonist was superior to the dual FPR1/FPR2 agonist in the lung. Recent work indicated a pathogenic contribution to pulmonary inflammation and fibrosis by FPR1 activation (Leslie et al, 2020): this may explain the lack of therapeutic efficacy of C43 on lung compliance and pulmonary immune cells. From a mechanistic point of view, macrophage subsets and infiltrating monocytes can impact on lung structure and provoke interstitial lung disease

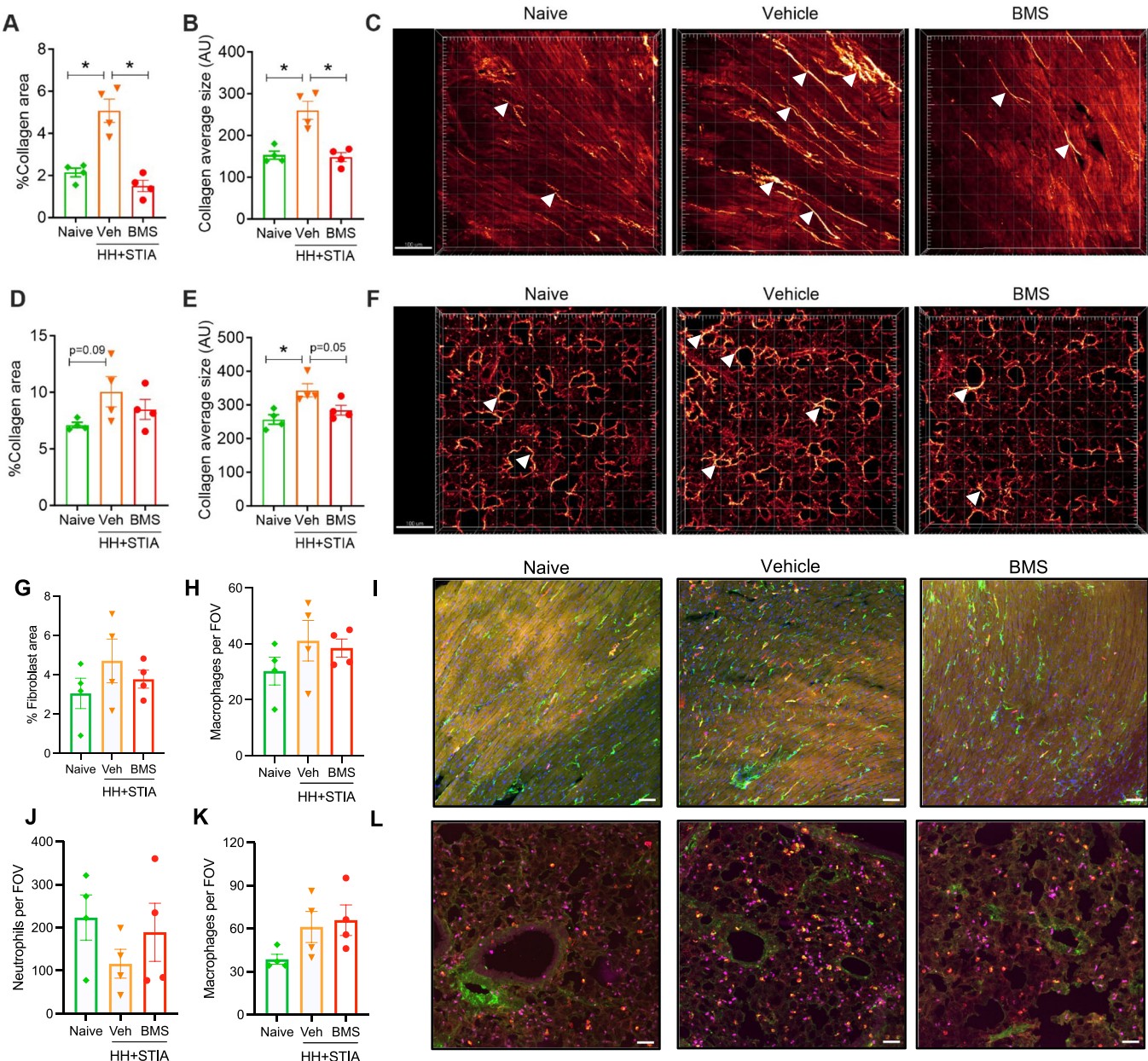

**Figure 7. Selective FPR2 agonism by BMS235 regulates collagen deposition in both heart and lung tissue of HH + STIA mice.**

HH + STIA was induced as in Fig. 1. From week 4, HH + STIA mice were treated with either vehicle or BMS235 (3 mg/kg per os) daily and analyses were conducted at week 6 (day 42). (A–F) Visualisation and analysis of collagen deposition in both hearts and lungs using second harmonic generation (SHG) by multiphoton microscopy. Data are mean ± SEM of $n = 4$ mice per group. *$P < 0.05$ vs. vehicle group. One-way ANOVA with Dunnett's multiple comparisons test (A) Percentage (%) of collagen area and (*, adjusted $P$ value = 0.0007 vs. naive; *, adjusted $P$ value = 0.0002 vs. BMS) (B) collagen average size (calculated by total collagen area divided by the number of collagen fibres) in the heart (*, adjusted $P$ value = 0.0012 vs. naive, *, adjusted $P$ value = 0.0009 vs. BMS). (C) Representative SHG images of the heart after conversion to maximum intensity projection (scale bars, 100 μm). (D) Percentage (%) of collagen area and (E) collagen average size in the lung (*, adjusted $P$ value = 0.0081 vs. naive). (F) Representative SHG images of the lung after conversion to maximum intensity projection (scale bars, 100 μm). Arrows highlight streaks or clusters of collagen fibres. (G–L) Visualisation and quantification of fibroblasts, macrophages and neutrophils in hearts and lungs using spinning disk confocal microscopy, after conversion to maximum intensity projection. Green: Vimentin, red: Galectin-3, purple: MRP14 (scale bars, 50 μm). Data are mean±SEM of $n = 4$ mice per group. (G) Percentage (%) of vimentin-positive fibroblast area and (H) galectin-3-positive macrophages per field of view in the heart. (I) Representative immunofluorescence spinning disc images of the heart. (J) Neutrophils and (K) macrophages per field of view in the lung. (L) Representative immunofluorescence spinning disc images of the lung. Source data are available online for this figure.

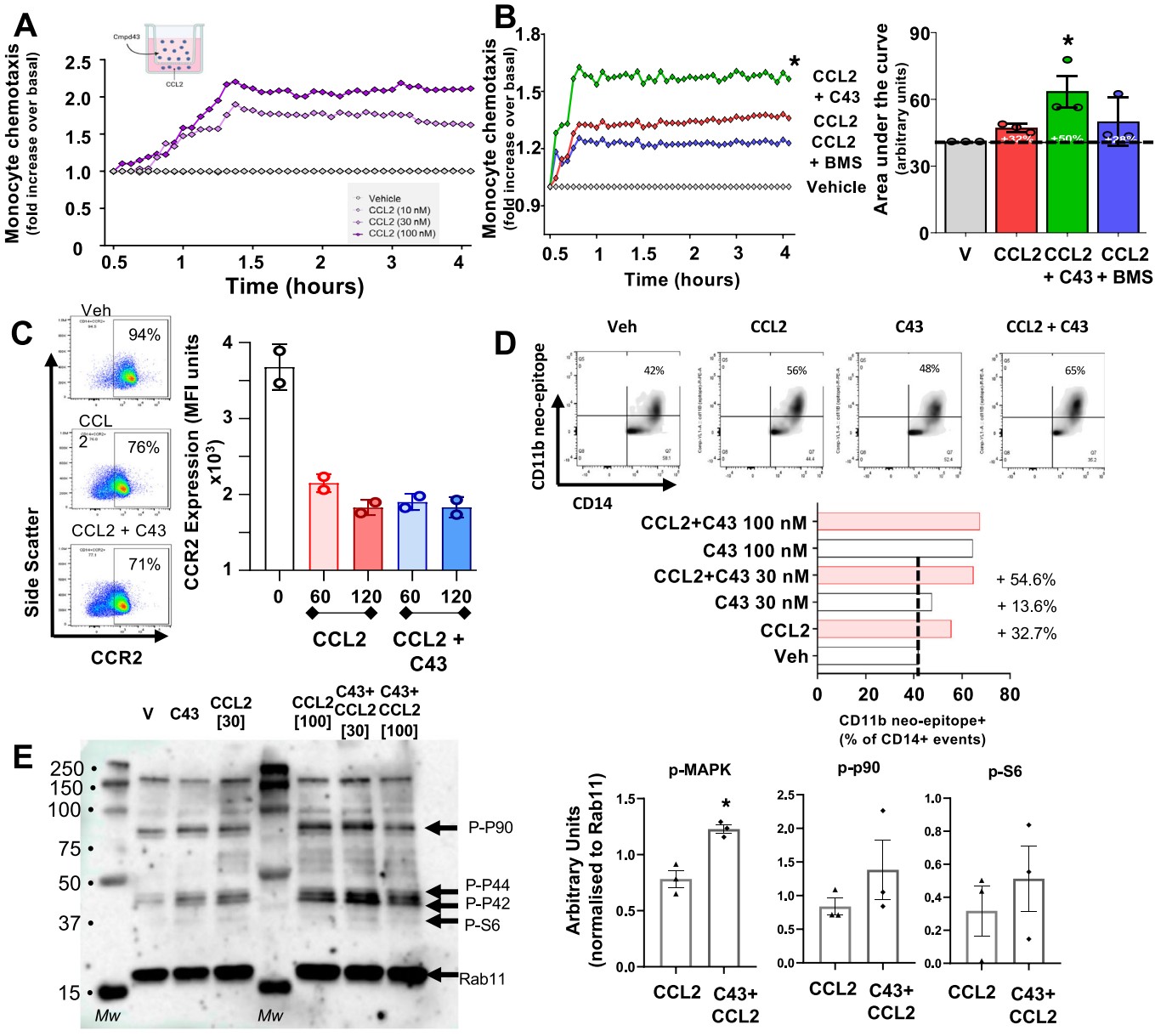

**Figure 8. C43 modulates human monocyte reactivity.**

(A) Chemotaxis of purified human peripheral blood monocytes was assessed using a xCELLigence™ DP system (see Appendix Methods). Representative concentration-response curves to CCL2. (B) CD14+ monocytes were incubated for 30 min with either vehicle, 100 nM C43 or 100 nM BMS235, prior to addition to the chemotactic chambers, with 30 nM CCL2 added to the lower chamber. Changes in impedance were recorded up to 4 h (*P value = 0.0402). (C) CD14+ monocytes were incubated with vehicle or 100 nM C43 for 30 min, prior to the addition of 30 nM CCL2 and CCR2 expression quantified by flow cytometry at 60 or 120 min, as indicated. Left: representative dot plots. Right: summary from experiments with two distinct cell donors. (D) CD14+ monocytes treated as in (C), though this time 10 nM CCL2 was used, while C43 was tested at two different concentrations. Expression levels of CD11b neo-epitope were quantified by flow cytometry. Top: representative density plots. Bottom: example of one experiment, indicating the additive effect of the lower C43 concentration (see Appendix Fig. S7B for experiments with three different cell donors). (E) PathScan™ Western blot phosphorylation screening on PBMCs treated with 30 nM C43 and two concentrations of CCL2. Left: representative blot. Right: cumulative data p44/42, p90 and S6 phosphorylation from experiments with three different cell preparations. See Appendix Fig. S8 for all three distinct blots. Data are mean ± SEM. *P < 0.05 vs. respective CCL2 value. One-way ANOVA with Tukey's multiple comparisons test. For (A, B): two-way ANOVA with Tukey's multiple comparisons test. For (E): unpaired two-tailed t test. Source data are available online for this figure.

(Min et al, 2023; Osterholzer et al, 2013; Rossaint et al, 2021). Our analyses revealed disturbances in the lung immune landscape, including dysregulation of Ly6C$^{hi}$ vs. Ly6C$^{lo}$ monocytes within the lung interstitium of HH + STIA mice, an effect paralleled by gene expression analysis as shown with nestin gene product expression

(Kadura and Raghu, 2021). Therapeutic efficacy through activation of FPR2 agonism counteracted the change in nestin while, again, the dual agonist was ineffective. Immune cell characterisation showed that C43, but not BMS235, augmented pro-inflammatory monocytes as well as CCR2 expression on F4/80$^+$

cells. In experiments with human cells, the dual FPR1/FPR2 agonist augmented monocyte recruitment in response to the monocyte chemoattractant CCL2. This potentiation was not dependent on the regulation of CCR2 expression, a conclusion aligned with previous work that showed lack of effect of FPR1 agonists on CCR2 recycling (Bednar et al, 2014). However, we identified signalling downstream of CCR2 where C43 intersects with CCL2, resulting in increased MAPK activation and higher integrin activation. The p44/42 MAPK-p90RSK (also termed RPS6KA1)-S6 pathway promotes cell chemotaxis and integrin activation (Abdulrahman et al, 2016; Li et al, 2007; Lian et al, 1999; Lin et al, 2015; Vial and McKeown-Longo, 2012) and was potentiated by C43 treatment of CCL2-activated monocytes. Phosphorylated p44/42 regulates monocyte locomotion (Huang et al, 2020; Malik et al, 2009) and this offers mechanistic explanation to the additive effect of C43. It was intriguing to observe that analyses with human monocytes also revealed direct chemotactic properties of C43, an effect reliant on both FPR1 and FPR2 as shown with selective antagonists, and in line with the pro-inflammatory actions of FPR1 agonists (Leslie et al, 2020). Therefore, higher pro-inflammatory monocyte numbers in the lung of HH + STIA mice treated with C43 can derive from an additive/synergistic action with CCL2 as well as from a direct pro-migratory effect of this small molecule.

These translational experiments with human monocytes were complemented with the assessment of BMS235 effects on human macrophages and fibroblasts. Indeed, an interesting regulation of macrophage-fibroblast crosstalk emerged with BMS235 reducing expression of selected markers of activation. These data resonate with the modulation of mouse fibroblast reactivity in the organs and tissues analysed, and are in agreement with original studies that reported FPR2 expression in synovial and cardiac fibroblasts and modulation of cellular reactivity upon receptor activation (Hashimoto et al, 2007; Qin et al, 2017). Finally, the indirect effect of FPR2 agonists on dermal fibroblasts has also been noted in settings of scleroderma (Park et al, 2019).

A dichotomy of behaviour was also observed with respect to joint disease. Only a selective FPR2 agonist alleviated arthritis symptoms, and positively impacted paw swelling and weight loss. This is apparently different from Kao et al, who reported anti-arthritic properties of C43 in settings of STIA (Kao et al, 2014): different time points (3-week arthritis here versus 9-days arthritis), doses (10 mg/kg here versus 30 mg/kg) and route of administration (oral here versus i.p.) can account for the different effects. Mechanistically, these observations were accompanied by the regulation of fibroblast and immune cell populations in the paw, including the reduction of Thy1$^+$FAP$^+$ immune effector fibroblasts implicated in severe and persistent arthritis (Croft et al, 2019). The reduction in joint-infiltrating neutrophils indicates that FPR2 agonists target inflammation-associated leukocyte subsets rather than the whole repertoire of immune cells. Neutrophils in the joint are implicated in both onset and persistence of STIA clinical scores (Chen et al, 2010) and presence and activation of neutrophil effector functions are targeted for barrier integrity and tissue homeostasis (Margraf and Perretti, 2022; Woodfin et al, 2016). Notably, BMS235 failed to modulate joint disease in K/BxN F1 mice, while affording protection against heart dysfunction, an observation that supports the engagement of mechanisms in a tissue- or organ-specific manner by formyl-peptide receptor agonists, as discussed further.

It remains unclear how HH + STIA leads to secondary organ damage, a feature not attained with STIA alone. Homocysteine interacts with the angiotensin II type I receptor (AT1), favouring AT1 receptor signalling and decreasing the EC$_{50}$ for angiotensin II: this can lead to aortic aneurysm formation in mice (Li et al, 2018). In addition, homocysteine might promote phenotypic changes in endothelial cells and smooth muscle cells, and this can be hypothesised since HH is directly connected to vascular calcification and acute coronary syndrome survival (Karger et al, 2020; Omland et al, 2000), as well as to obstructive pulmonary diseases (Nunomiya et al, 2013; Seemungal et al, 2007). The specific cellular target altered by hyper-homocysteine in normal or arthritic mice remains to be identified but involvement of the vasculature remains plausible (Yuan et al, 2022). Further studies are required to delineate this pathogenic mechanism.

The experimental data discussed so far are in line with (i) evidence that in global *Fpr2* null mice failure to activate this receptor leads to cardiometabolic syndrome, diastolic dysfunction and reduced survival rates with signs of multi-organ non-resolving inflammation (Tourki et al, 2020) and (ii) genomic profiling of RA patient blood. Data mining of published databases allowed the identification of increased FPR2 gene expression in RA PBMC: one could speculate the body itself might try to trigger necessary resolution responses when exaggerated inflammation is sensed, like during active arthritis. Failure to start resolution mechanisms might be due to limited availability of ligand-receptor interactions, a shortfall which might be corrected by exogenous delivery of FPR2 agonists.

There are at least two further lines of investigation that deserve attention. First, further elucidation of pathological mechanisms is required for secondary organ dysfunction during RA. The current study offers initial indications of what might become a series of disease subsets associated with organ-dependent cellular changes which can be targeted in a specific manner. Second, it is possible the beneficial pharmacology of FPR agonists may be enhanced in combination with other anti-arthritic therapies, with protection against secondary organ damage to become a key signature for this class of compounds. To this end, it is worth noting that FPR2 cell surface expression was unchanged even after two weeks of compound administration; the lack of prolonged disappearance of the receptor could be due to the rapid recycling promoted by BMS235 application, as shown also with primary human cardiac fibroblasts (Lupisella et al, 2022). Such molecular pharmacology of BMS235 is different from that of other agonists which have been tested clinically (Stalder et al, 2017).

To conclude, in settings of inflammatory arthritis, cardiac injury is independent of the joint since BMS235, but not C43 controls joint disease, while both attenuate cardiac dysfunction. At variance from the heart, the defect in lung function can be linked to the joint, as BMS235 protected both tissues, whereas C43 failed to do so. Nonetheless, distinct cellular effects are evoked by BMS235 in lung *versus* joint, and selective FPR2 agonists are superior to dual FPR1/FPR2 activation. This study paves the way to therapeutic strategies centred on FPR2 to mitigate, if not prevent, morbidity and mortality events experienced by RA patients.

# Methods

**Reagents and tools table**

| Reagent/resource | Reference or source | Identifier or catalogue number | |
|---|---|---|---|
| **Experimental models** | | | |
| HH | Zhou et al 2001; https://doi.org/10.1161/hq0901.096582 | | |
| STIA | Garrido-Mesa et al, 2022 https://doi.org/10.3389/fimmu.2022.1078678 | | |
| KBxN F1 | See references Chen et al, 2021 Kouskoff et al, 1996 Matsumoto et al, 1999 in the main text | | |
| NOD/ShiLtJ | Charles River | | |
| KRN | See Kouskoff et al, 1996 Matsumoto et al, 1999 in the main text | | |
| C57BL/6J | Charles River | | |
| Human cardiac fibroblasts | Promo-cell | C-12375 | |
| **Recombinant DNA** | | | |
| **Antibodies** | | **Clone/dilution** | **Concentration** |
| CD45 | Biolegend | 30-F11 (1:200) | 0.2 mg/mL |
| Ly-6B.2 | Abcam | 7/4 (1:60) | 1.0 mg/mL |
| Ly6G | Biolegend | 1A8 (1:60) | 0.2 mg/mL |
| CD54 | Biolegend | YN1/1.7.4 (1:100) | 0.2 mg/mL |
| CXCR1 | R&D Systems | 1122 A (1:60) | 0.2 mg/mL |
| CD11b | Biolegend | M1/70 (1:200) | 0.5 mg/mL |
| CD62L | Biolegend | MEL-14 (1:100) | 0.2 mg/mL |
| CD62L | Biolegend | MEL-14 (1:200) | 0.5 mg/mL |
| CD62P | BD Biosciences | RB40.34 (1:60) | 0.5 mg/mL |
| CD41 | Biolegend | MWReg30 (1:100) | 0.5 mg/mL |
| CD41 | eBioscience | eBioMWReg30 (1:60) | 0.2 mg/mL |
| GPIb | Emfret | Xia.G5 (1:30) | Not provided |
| GPVI | Emfret | JAQ1 (1:10) | Not provided |
| GPIIb/IIIa (activated) | Emfret | JON/A (1:10) | Not provided |
| FPR2 | Novusbio | Polyclonal (1:100) | 0.56 mg/mL |
| FPR2 | Biorbyt | Polyclonal (1:40) | 0.5 mg/mL |
| FPR2 | Santa Cruz Biotechnologies | Clone GM1B6 (1:40) | 0.2 mg/mL |
| Biotin Conjugation Lightning-Link kit | Abcam | ab201795 | Not provided |
| Streptavidin in AF488 | Biolegend | S32354 | 2 mg/mL |
| F4/80 | Biolegend | BM8 (1:30) | 0.2 mg/mL |
| CX3CR1 | Biolegend | SA011F11 (1:30) | 0.5 mg/mL |
| MHCII | eBioscience | M5/114.15.2 (1:25) | 0.2 mg/mL |

| Reagent/resource | Reference or source | Identifier or catalogue number | |
|---|---|---|---|
| SiglecF | MACS Miltenyi Biotec | ES22-10D8 (1:60) | 0.15 µg/mL |
| CD11c | eBioscience | N418 (1:25) | 0.2 mg/mL |
| CD206 | Biolegend | C068C2 (1:60) | 0.1 mg/mL |
| MerTK | Biolegend | 2B10C42 (1:60) | 0.2 mg/mL |
| F4/80 | Biolegend | BM8 (1:100) | 0.2 mg/mL |
| CD14 | Biolegend | Sa14-2 (1:60) | 0.2 mg/mL |
| CD3 | Biolegend | 17A2 (1:100) | 0.5 mg/mL |
| CD4 | Biolegend | RM4-4 (1:400) | 0.2 mg/mL |
| CD8 | Biolegend | 53-6.7 (1:100) | 0.5 mg/mL |
| CD25 | Biolegend | PC61 (1:100) | 0.2 mg/mL |
| E780 Viability stain | ThermoFisher | 65-0865-14 (1:1000) | Not provided |
| CD80 | Biolegend | 16-10A1 (1:60) | 0.2 mg/mL |
| CD45 | Macs Miltenyi Biotec | REA737 (1:60) | 0.15 mg/mL |
| Ly6C | Biolegend | HK1.4 (1:60) | 0.2 mg/mL |
| CD40 | Biolegend | 3/23 (1:60) | 0.5 mg/mL |
| CD40L | Biolegend | MR1 (1:60) | 0.2 mg/mL |
| Fc Block | Biolegend | Trustain FcX (1:200) | 0.5 mg/mL |
| CD44 | Biolegend | IM7 (1:100) | 0.2 mg/mL |
| Lineage | Biolegend | N/A (1:200) | Not provided |
| CD31 | BD Biosciences | MEC 13.3 (1:200) | 0.5 mg/mL |
| Thy1.2 | Biolegend | 30-H12 (1:100) | 0.5 mg/mL |
| CD34 | Biolegend | HM34 (1:100) | 0.2 mg/mL |
| Podoplanin | Biolegend | 8.1.1 (1:100) | 0.2 mg/mL |
| Cardiac Troponin T | BD Biosciences | 13-11 (1:100) | 0.2 mg/mL |
| MEFSK4 | Macs Miltenyi Biotec | mEF-SK4 (1:20) | 0.15 mg/mL |
| CD31 | Biolegend | 390 (1:250) | 0.2 mg/mL |
| FAP-alpha | Abcam | Polyclonal (1:100) | 1.0 mg/mL |
| Goat anti-Rabbit IgG (H + L) Secondary Ab | Invitrogen | Polyclonal (1:100) | 0.8 mg/mL |
| Ly6G | BD Biosciences | 1A8 (1:100) | 0.2 mg/mL |
| CD45 | BD Biosciences | 30-F11 (1:100) | 0.2 mg/mL |
| CD11b | eBioscience/Invitrogen | M1/70 (1:200) | 0.2 mg/mL |
| CD115 | Biolegend | AFS98 (1:100) | 0.2 mg/mL |
| CD43 | BD Biosciences | S7 (1:100) | 0.2 mg/mL |
| Siglec-F | BD Biosciences | E50-2440 (1:200) | 0.2 mg/mL |
| Ly6C | eBioscience | HK1.4 (1:200) | 0.2 mg/mL |
| CCR2 | BD Biosciences | 475301 (1:200) | 0.2 mg/mL |
| CD64 | Biolegend | X54-5/7.1 (1:100) | 0.2 mg/mL |
| Lineage | Biolegend | 133313 (1:200) | Not provided |
| Epcam | Biolegend | G8.8 (1:100) | 0.5 mg/mL |
| PDGR1 | Biolegend | APA5 (1:100) | 0.2 mg/mL |
| MRP14 | Abcam | 2B10 (1:400) | 10 µg/mL |
| Gal-3 | R&D | Polyclonal (1:100) | 0.2 mg/mL |
| Vimentin | ThermoFisher Scientific | 6K21 (1:100) | 1.0 mg/mL |
| Donkey anti-goat | Invitrogen | Polyclonal (1:400) | 2 mg/mL |

| Reagent/resource | Reference or source | Identifier or catalogue number | |
|---|---|---|---|
| PathScan Western blot antibody | Cell Signaling | (1:1000) | Not provided. |
| **Oligonucleotides and other sequence-based reagents** | | | |
| qPCR primers | | | |
| **mNe tin** | | | |
| Forward | TAAAAGCTCCAAGGGCCACTC | | |
| Reverse | GATTCTTCCCCGACGCAACC | | |
| **mSPP1** | | | |
| Forward | CTGGCTGAATTCTGAGGGACTA | | |
| Reverse | TGAGATGGGTCAGGCACCA | | |
| **mFra-2** | | | |
| Forward | ACGCTCACATCCCTACAGTC | | |
| Reverse | CGGATTCGACGCTTCTCCT | | |
| mCSF1R primer | Qiagen | QT01055810 | |
| m-TGF-b | Qiagen | QT00145250 | |
| mFPR2 mIL-6 **mCol1A1** Forward: ACGAGTCACACCGGAACTTG Reverse: TGGGGTGGAGGGAGTTTACA | Qiagen | QT00171514 QT00098875 | |
| Human 18S | Qiagen | QT00199367 | |
| Human CCL2 | Qiagen | QT00212730 | |
| Human IL8 | Qiagen | QT00000322 | |
| **Chemicals, enzymes and other reagents** | | | |
| D,L-homocysteine | Sigma-Aldrich | | |
| Liberase TH | Sigma-Aldrich | | |
| DNase1 | Sigma-Aldrich | | |
| Hyaluronidase Type 1-S | Sigma-Aldrich | | |
| Collagenase A | Sigma-Aldrich | | |
| Histopaque 1077 | Sigma-Aldrich | | |
| CD14 positive selection kit | Miltenyi Biotech | | |
| C5a | R&D | | |
| Cyclosporin H | Alexis Biochemicals | | |
| WRW4 | Bachem | | |
| HALT phosphatase inhibitors | Thermo Scientific | | |
| Immobilon Forte Western HRP Substrate | Millipore | | |
| **Software** | | | |
| IAAA database | galaxy.ustc.edu.cn/IAAA/dataset/bulkRNA/ | | |
| LabChart v8 | ADInstruments | | |
| FlowJo V10 | Becton, Dickinson & Company | | |
| Bio-Plex Manager | Bio-Rad | | |
| ImageJ | ImageJ.net | | |
| GraphPad PRISM v8 | GraphPad Software | | |
| Imaris | Oxford Instruments | | |

| Reagent/resource | Reference or source | Identifier or catalogue number |
|---|---|---|
| **Other materials** | | |
| C43, Agonist | Merck Millipore | |
| BMS986235, Agonist | Bristol Myers Squibb | |
| Vevo-3100 imaging system | Fujifilm Visualsonics | |
| ProCyte | Idexx Europe B.V., Netherlands | |
| CODA blood pressure monitor | Kent Scientific | |
| AttuneNxT | ThermoFisher | |
| BD FACSymphony | BD | |
| BD LSRFortessa | BD | |
| Mouse creatinine Assay Kit | Crystal Chem High Performance Assays, elk Grove, USA | |
| AST-ELISA | Abcam | Ab263882 |
| Luminex panels | Millipore Sigma, Bio-Techne | MHSTCMAG-70K, MCYTOMAG-70K, MECY2MAG-73K, MTH17MAG-47K |
| FLEXMAP 3D instrument | Luminex ThermoFisher Scientific | |
| Homocysteine ELISA | ELK biotechnology | ELK8718 |
| Precellys Lysing kit | Bertin Technologies | |
| Precellys homogenizer | Bertin Technologies | |
| RNeasy Fibrous Tissue mini kit | Qiagen | |
| Nanodrop 2000 | ThermoFisher | |
| RevertAid First Strand cDNA synthesis kit | ThermoFisher | |
| QuantStudio 5 RT PCR, 384 well block | ThermoFisher | |
| Primers | Qiagen (Hprt1), Merck (Nestin) | |
| xCELLigence DP system | Agilent | |
| Attune Cytpix | Invitrogen | |
| Azure 400 CCD system | Azure Biosystems | |
| Multiphoton laser scanning microscope | Leica | |
| Nikon CSU-W1 SoRa spinning disk confocal microscope | Nikon | |
| **Databases** | | |
| Database/Dataset | | URL/Access |
| IAAA database | | galaxy.ustc.edu.cn/IAAA/dataset/bulkRNA/ |
| Gene Expression Omnibus dataset for GSE90081 | | https://www.ncbi.nlm.nih.gov/geo/query/acc.cgi?acc=GSE90081 |
| Gene Expression Omnibus dataset for GSE66573 | | https://www.ncbi.nlm.nih.gov/geo/query/acc.cgi?acc=GSE66573 |
| Gene Expression Omnibus dataset for GSE66763 | | https://www.ncbi.nlm.nih.gov/geo/query/acc.cgi?acc=GSE66763 |
| Gene Expression Omnibus dataset for GSE77598 | | https://www.ncbi.nlm.nih.gov/geo/query/acc.cgi?acc=GSE77598 |
| Gene Expression Omnibus dataset for GSE112057 | | https://www.ncbi.nlm.nih.gov/geo/query/acc.cgi?acc=GSE112057 |
| Gene Expression Omnibus dataset for GSE122459 | | https://www.ncbi.nlm.nih.gov/geo/query/acc.cgi?acc=GSE122459 |
| Gene Expression Omnibus dataset for GSE123786 | | https://www.ncbi.nlm.nih.gov/geo/query/acc.cgi?acc=GSE123786 |
| Gene Expression Omnibus dataset for GSE125977 | | https://www.ncbi.nlm.nih.gov/geo/query/acc.cgi?acc=GSE125977 |

| Reagent/resource | Reference or source | Identifier or catalogue number | | |
|---|---|---|---|---|
| **Author year dataset title dataset URL database and identifier** | | | | |
| **Author(s)** | **Year** | **Dataset Title** | **Dataset URL** | **Database and Identifier** |
| Shchetynsky K et al | 2017 | Discovery of new candidate genes for rheumatoid arthritis through integration of genetic association data with expression pathway analysis | https://www.ncbi.nlm.nih.gov/geo/query/acc.cgi?acc=GSE90081 | GEO GSE90081 |
| Spurlock III CF et al | 2015 | Defective structural RNA processing in relapsing-remitting multiple sclerosis | https://www.ncbi.nlm.nih.gov/geo/query/acc.cgi?acc=GSE66573 | GEO GSE66573 |
| Cao Y et al | 2015 | Functional Inflammatory Profiles Distinguish Myelin-Reactive T Cells from Patients with Multiple Sclerosis | https://www.ncbi.nlm.nih.gov/geo/query/acc.cgi?acc=GSE66763 | GEO GSE66763 |
| Binder MD et al | 2016 | Expression of MERTK based on Multiple Sclerosis (MS) risk haplotype | https://www.ncbi.nlm.nih.gov/geo/query/acc.cgi?acc=GSE77598 | GEO GSE77598 |
| Mo A et al | 2018 | Whole Blood Transcriptome Profiling in Juvenile Idiopathic Arthritis and Inflammatory Bowel Disease | https://www.ncbi.nlm.nih.gov/geo/query/acc.cgi?acc=GSE112057 | GEO GSE112057 |
| Tokuyama M et al | 2018 | SLE PBMC RNA-seq | https://www.ncbi.nlm.nih.gov/geo/query/acc.cgi?acc=GSE122459 | GEO GSE122459 |
| Catapano M et al | 2020 | Whole blood RNAseq from Generalised Pustular Psoriasis patients and healthy individuals | https://www.ncbi.nlm.nih.gov/geo/query/acc.cgi?acc=GSE123786 | GEO GSE123786 |
| Parkes JE et al | 2020 | MicroRNA and mRNA profiling in the idiopathic inflammatory myopathies | https://www.ncbi.nlm.nih.gov/geo/query/acc.cgi?acc=GSE125977 | GEO GSE125977 |

## Study design

Sample sizes were selected based on previous experience with similar phenotypic models. Animals were randomised and experiments performed in a blinded manner. Animals were age and sex-matched. Details on sample sizes representing biological replicates and statistical tests are detailed in figure legends and in the Statistical Analysis section of 'Methods'.

The local 'Animal Use and Care Committee' approved animal experiments in accordance with the derivatives of both, the 'Home Office guidance on the Operation of Animals (Scientific Procedures) Act 1986', and the 'Guide for the Care and Use of Laboratory Animals' of the National Research Council. Ethics approval no. is PA672E0EE; Queen Mary Research ethics committee 3.

## Animals and reagents

All animal experiments were carried out in accordance with the institutional Animal Welfare Ethical Review Body and UK Home Office guidelines. For serum collection, K/BxN F1 arthritic mice were obtained from mating NOD/ShiLtJ mice (Charles River) and KRN T-cell receptor transgenic mice, both on a C57BL/6 J background. This mouse colony was originally described by Mathis and Benoist

(Kouskoff et al, 1996; Matsumoto et al, 1999). Animals were bred in house, develop arthritis from 4 to 5 weeks of age and are routinely culled at weeks 14–15 to prepare the serum which is stored at −80 °C prior to further use. Male C57BL/6 J mice (8-week-old) were acquired from Charles River UK. All mice were maintained on a standard chow pellet diet (PicoLab™ Rodent Diet 20' LabDiet) with *ad libitum* access to standard water until start of experiment. In the diet, specific components had the following proportions: cystine 0.36%, methionine 0.62%, folic acid 3 ppm, vitamin B12 51 µg/kg. Values of these nutrients are below the levels reported to impact circulating homocysteine concentrations (Brutting et al, 2021).

## Model of hyper-homocysteinemia and serum-transfer-induced arthritis (HH + STIA)

D,L-homocysteine (Sigma-Aldrich) was dissolved (1.8 g/L) and administered via drinking water for 3 weeks to C57BL/6 J mice. After the homocysteine-induction period, arthritogenic serum (100 µl) was injected i.p., at days 21, 23, 28, 30, 35 and 37. Arthritis was monitored for 3 weeks, with mice continuing to access homocysteine in the drinking water (total duration of 6 weeks; see Fig. 1A). Several analyses were conducted, as detailed below, in a blinded fashion.

Two experimental analyses were conducted: first, HH + STIA was compared with STIA alone and HH alone; second, therapeutic intervention was tested in HH + STIA starting after arthritis induction (week 4) for an additional 2 weeks with daily oral administration of (a) vehicle (0.5% CMC at 5 ml/kg); (b) BMS986235 (abbreviated to BMS235 3 mg/kg p.o) (Asahina et al, 2020) at; (c) compound 43 (C43; 10 mg/kg p.o) (Burli et al, 2006).

## Macroscopic analyses in the HH + STIA model

### Assessment of arthritis scores

Progression of joint disease was monitored by assessment of clinical scores (maximum 12 points) with up to 3 points for each limb (0: no inflammation, 1: inflammation visible in either individual phalanges, localised wrist/ankle or surface of paw; 2: combined inflammation in two modalities of the paw; 3: major swelling and inflammation in all modalities of the paw). Mice were scored daily, and plethysmography performed using a hydro-plethysmometer (Ugo Basile, Milan, Italy), as reported (Montero-Melendez et al, 2014).

### Organ analysis and bronchoalveolar lavage

At the indicated time points, mice were sacrificed, and blood was removed via puncture of the cava vein. Lungs were lavaged to obtain bronchoalveolar lavage (BAL) via intratracheal tube placement using a sterile filtered physiologic saline solution (0.9% NaCl). Animals were then perfused with ice-cold PBS (40 ml) *via* cardiac puncture until colour of organs cleared. The heart was then perfused again via the left ventricle and the aorta clamped to ensure perfusion of coronaries. Lungs and hearts were removed and weighed prior to dissociation. Organs were cut, digested (lung: collagenase A, hyaluronidase type 1-S, and DNase1, 45 min, all from Sigma-Aldrich; heart: Liberase TH and DNase1, 30 min, from Sigma-Aldrich) at 37 °C, passed through a 70-μm cell-strainer followed by a 40-μm cell-strainer and subsequently stained for flow cytometry with indicated antibodies. Paws were collected, skin and fur carefully removed, and digested (collagenase D, from Sigma-Aldrich and DNAse1, from Roche, two times 30 min with agitation) at 37 °C, passed through a 70-μm cell-strainer and subsequently stained for flow cytometry with indicated antibodies.

### Assessment of pulmonary compliance

Lung compliance was assessed as described previously (Rossaint et al, 2021). Briefly, an intratracheal tube was inserted via tracheotomy and a precision syringe coupled to a pressure gauge was connected. Static lung compliance was measured by assessment of the required pressure to inflate the lung by an inspiratory volume of 1 ml room air and calculated in mbar/ml.

### Assessment of cardiac function (echocardiography)

M-mode and Doppler echocardiography were performed in all groups at week 3, 4 and 6. Animals were subjected to isoflurane anaesthesia (3% for induction and 0.5 to 0.7% for the duration of the procedure). Before assessment of cardiac function, fur was removed from the chest area to allow accurate assessment of cardiac function. Body temperature was maintained at 36.5-37.5 °C. Heart rate was constantly measured via ECG. A Vevo-3100 imaging system with a MX550D 40 MHz linear probe (VisualSonics, Toronto, Canada) was used for echocardiography analyses. Diastolic transmitral left ventricle (LV) inflow images were

acquired from apical four-chamber views using pulsed-wave Doppler. The sample volume was positioned at the tips of mitral valve leaflet in the mitral valve annulus, the ultrasound beam was in parallel with the direction of blood flow to record maximal transmitral flow velocities. Early (E) and late (atrial, A) peak filling blood flow velocities and E-wave deceleration time were calculated. The E/A ratio represents the ratio of E wave to A wave. Left atrial area was measured in apical four-chamber views, borders of the left atrium were traced just before mitral valve opening at end ventricular systole. M-mode echocardiography was performed to measure interventricular septum thickness, left ventricle internal dimension in diastolic phase, and percentage ejection fraction in the parasternal short axis view at the level of the papillary muscles (Chen et al, 2021).

### Measurement of blood cell counts

Whole blood was collected 24 h after last dosing into EDTA-coated tubes and subjected to haematology analysis using a ProCyte (Idexx Europe B.V., the Netherlands) automated haematology analyser according to the manufacturers' instructions.

### Measurement of blood pressure

A CODA blood pressure monitor was used, and measurements recorded using LabChart v8 software according to manufacturers' recommendations and as described previously (Gee et al, 2022). In brief, animals were warmed at 37 °C for 5 min in animal heating boxes and placed on thermal pads during measurements to maintain body temperature. To record blood pressure, a tail cuff pressure sensor for mice was used and animals were immobilised for the duration of measurements in a mouse restrainer. Recordings were performed following an initial training period of the mice until they were accustomed to restrainers and cuff placement.

## Imaging of mouse hearts and lungs

### Sample preparation

Mice were terminally anaesthetised with xylazine (7.5 mg/kg) and ketamine (150 mg/kg), blood was collected from the inferior vena cava into heparin-coated syringes, and plasma was stored at -80 °C. The descending aorta and brachial artery were clamped, and lungs were perfusion-fixed with 5 ml of PBS followed by 5 ml of 2% PFA in PBS at 2.5 ml/min using a syringe pump (Harvard apparatus). The upper trachea was exposed and a 22 G plastic catheter was inserted and ligated into position for lung inflation-fixation for ~10 min with 1:1 ratio of 4% PFA and optimal cutting temperature medium (OCT). Hearts and lungs were ligated, excised and immersed in 4% PFA for 2 h and then transferred into 10%, 15%, and 30% sucrose for 2 h, overnight and 4 h, respectively at 4 °C. Hearts were then cut coronally so that adjacent tissue sections from the same heart sample were used for second harmonic generation (SHG) imaging and immunofluorescence. Tissues were embedded and frozen in OCT on pre-cooled isopropanol, and 30-μm frozen sections were cut using a Bright OFT5000 cryostat and stored at −80 °C prior to subsequent analysis.

### Second harmonic generation (SHG) imaging and collagen quantification

SHG imaging was performed for label-free imaging of collagen on 30 μm frozen sections of heart and lung tissue. Images were

acquired with a SP8 DIVE multiphoton (MP) laser scanning microscope (Leica) with 25× water immersion motCORR objective (Leica). SHG signals were collected at 450 nm after excitation at 900 nm with an Insight Duals-067 MP laser (Spectra-Physics). XY axes acquisition was performed with 0.75 zoom factor, a speed of 600 Hz bidirectional scan at $1024 \times 1024$ pixel resolution, and under four times line average, while Z axes was set at 1.5 μm Z-step size. Five random images per lung section and six random images per heart section (two at interventricular septum, two at left ventricular apex and two at left ventricular wall) were acquired.

Quantification of collagen contents was performed on SHG images after conversion to maximum intensity projection (ImageJ), then segmented using the machine learning Ilastik software, and quantified via a macro on ImageJ. Representative images and videos were generated using Imaris (Oxford Instrument).

### Immunofluorescence imaging

Tissue sections (30 μm) were incubated in PBS with 2% BSA and 0.1% Triton X-100 for 30 min to block and permeabilize the tissue. Primary antibodies were diluted in block/perm buffer to the concentrations specified below and incubated at room temperature for 1 h. Sections were washed three times in PBS, and secondary antibody was incubated for 1 h at room temperature. After washing, sections were counterstained with DAPI and mounted with ProLong Gold Antifade Mountant. Sections were left to cure overnight before imaging and analysis. Images were acquired using a Nikon CSU-W1 SoRa spinning disk confocal microscope with a 20×, 0.8 numerical aperture objective lens. Three random regions per lung section, and four regions for left ventricle sections, were selected for imaging. Images were captured as Z-stacks with a step size of 1 μm. For analysis, each Z-stack was transformed into a maximum intensity projection image (ImageJ), background was subtracted, and a Gaussian filter was applied. Channel thresholding was applied to accurately map cells within the image and MRP14$^+$ neutrophils and Galectin-3$^+$ macrophages were quantified per field of view (FOV). Due to the dense and overlapping cellular distribution of vimentin $^{ve+}$ cardiac fibroblasts, the percentage area covered by the cells was calculated. All images from each animal were averaged to generate a single data point per animal.

## Cellular analyses for the HH + STIA model

### Flow cytometry measurements

Samples were stained using indicated antibodies and measured on an AttuneNxT (ThermoFisher) (lung and blood), a BD FAC-Symphony (paws) and a BD LSRFortessa (heart) flow cytometer (BD Biosciences, San Jose, California, USA). Single-cell gating strategy was applied in all experiments except when cellular aggregates were to be included and/or for specific cellular aggregate detection. Neutrophils were identified as CD45+7/4+ and/or Ly6G+. Macrophage subsets were determined according to the described characteristics. Utilised antibodies are listed in Appendix Table S1. Multicolour counting beads as volumetric reference were from BD Biosciences. Data were analysed using FlowJo analysis software (Treestar Inc., Ashland, USA).

## Soluble mediator quantification

Creatinine levels were determined using an enzymatic detection method (Mouse Creatinine Assay Kit; Crystal Chem High-

Performance Assays; Elk Grove Village, IL, USA). For the assessment of circulating AST levels, an AST-ELISA assay (Abcam ab263882) was performed according to the manufacturer's instructions. The concentration of additional soluble protein analytes was measured using a Luminex™ Bead-Based multiplexed assay with 4 commercially available panels (MHSTCMAG-70K, MCYTOMAG-70K, MECY2MAG-73K, MTH17MAG-47K; Millipore Sigma) and custom developed analyte panels (Bio-Techne). Data were collected using a FLEXMAP 3D™ instrument. Standard curves were derived and analyte concentrations were calculated with Bio-Plex Manager™ (Bio-Rad) software using phycoerythrin median fluorescence intensity (MFI) unit data. Samples were diluted according to the manufacturers' instructions and pre-tests with the aim to bring particular analytes within the dynamic range of the assay.

At the end of the experiment, plasma samples were used to quantify homocysteine concentrations, with values of $2384 \pm 794$ μM in the HH and $4544 \pm 998$ in the HH + STIA groups ($n = 7$ in both cases, 3 female and 4 male mice). In KBxN F1 mice, plasma homocysteine levels were $934 \pm 254$, whereas control KRN mouse plasma contained $226 \pm 38$ μM. The latter values are within the norm for different mouse genotypes (Ernest et al, 2005); Plasma homocysteine was quantified by ELISA (Cat: ELK8718; ELK biotechnology, Denver, USA).

## Molecular analyses

### qPCR of tissue samples

Lungs were flushed intratracheally and *via* cardiac perfusion until cleared and subsequently homogenised using a Precellys Lysing kit on a Precellys homogenizer. RNA extraction was performed using RNeasy Fibrous Tissue Mini Kit. RNA concentration and purity were measured on a Nanodrop2000 (ThermoFisher). A RevertAid First Strand cDNA Synthesis Kit (ThermoFisher) was applied for cDNA synthesis. A QuantStudio 5 RT PCR with a 384 well block was used for qPCR. *Hprt1* was used as reference gene in all samples (Fragoulis et al, 2021; Norman et al, 2022). Ct values were extracted for further calculations and fold change in expression relative to housekeeping reference gene assessed. Primers were from Qiagen (*Hprt1*) and Merck (*Nestin*), respectively.

## K/BxN F1 model of spontaneous arthritis

A number of selected analyses were repeated with the K/BxN F1 mouse colony described above. Herein, male mice ($n = 5$ per group) were treated with vehicle or BMS235 (3 mg/kg p.o) from week 4 of age, when arthritis is fully developed (Chen et al, 2021). Compounds were administered daily via oral gavage for four weeks and cardiac analyses conducted at week 8 of age by echocardiography, as detailed above.

## Human sample analyses

### Inquiry of human bulk RNAseq datasets

Data were accessed and processed as described previously (Shen et al, 2022). In brief, data pre-processing contained in this pipeline included Trimmomatic, STAR and DESeq2 (Bolger et al, 2014; Dobin et al, 2013). Datasets for "healthy" and "RA" were chosen for direct comparison. Volcano plot for expression analysis is presented.

## Monocyte chemotaxis

Experiments with human cells were approved by Queen Mary Ethics Committee (QME24.0391; 'The role of healthy human leukocytes in inflammation'). Informed consent was obtained from all volunteers; the experiments conformed to the principles set out in the WMA Declaration of Helsinki and the Department of Health and Human Services Belmont Report. Peripheral blood mononuclear cells (PBMCs) were isolated from human blood using Histopaque 1077 (Sigma-Aldrich, Missouri, USA) and CD14$^+$ cells positively selected with CD14 MicroBeads (Miltenyi Biotech, Bergisch Gladbach, Germany). CD14$^+$ monocytes ($0.25 \times 10^6$) were treated for 30 min with vehicle, 100 nM C43 or 100 nM BMS235 prior to transfer to the top chamber of a preloaded cell invasion and migration plate (CIM-Plate-16) of the xCELLigence™ DP system (Agilent, California, USA). The chemokine (C-C motif) ligand 2 (CCL2) (usually at 30 nM; 10–100 nM in concentration-response experiments) was added to the bottom of the plate to assess the real-time chemotaxis by changes in movement (impedance). In some experiments, C5a (10 nM; R&D) was used. In other experiments, C43 was added to the bottom of the plate, with CD14$^+$ monocytes incubated with vehicle or cyclosporin H (CyH; 1 µM; Alexis Biochemicals) or WRW4 (1 µM; Bachem) for 30 min prior to transfer onto the xCELLigence™ DP system. In all cases, cell migration was followed up to 4 h, since it plateaued beyond this time point. Experiments were conducted with distinct CD14$^+$ monocyte donors, run with three technical replicates.

## CCR2 expression by flow cytometry

Briefly, purified CD14$^+$ monocytes ($5 \times 10^5$ cells) were incubated with vehicle or C43 (100 nM) for 30 min at 37 °C. Then, CCL2 (100 nM) was added and incubation carried for 60 or 120 min. At the end, cells were stained with monoclonal antibodies specific for CD14 (BioLegend, APC-Cy7, clone HCD14, 1:100) and CCR2 (BD Biosciences, BV-786, clone LS132.1D9, 1:100) for 30 min at 4 °C. All data were collected on Attune CytPix (Invitrogen) for further analysis using FlowJo software.

## CD11b activation epitope

Histopaque-purified PBMCs, and not monocytes, were used to minimise cell activation. Cells ($5 \times 10^5$) were incubated with C43 (10–100 nM) or vehicle for 30 min at 37 °C. Then, anti-CD11b (eBioscience, PE, clone CBRM 1/5, 1 µl/well) and CCL2 (30 nM) were added and incubation carried for further 30 min. At the end, cells were stained with anti-CD14 (Invitrogen, APC, clone 61D3, 1:100) for 30 min at 4 °C. Samples were acquired on Attune CytPix (Invitrogen) and analysed using FlowJo software.

## PathScan western blot

Histopaque-purified PBMCs ($5 \times 10^5$) were used to minimise cell activation. Cells were stimulated as indicated and immediately lysed using RIPA lysis buffer with HALT phosphatase inhibitors (Thermo Scientific). Lysates were run using 10% polyacrylamide gels. Membranes were incubated with PathScan antibody (Cell Signaling; 1:1000) overnight at 4 °C. The secondary antibody (1:5000) was incubated for 1 h at room temperature. An Immobilon Forte Western HRP Substrate (Millipore) kit was used for development, and an Azure 400 western blot CCD imaging system for image acquisition. Quantification was performed using ImageJ.

## The paper explained

### Problem

Patients affected by autoimmune diseases experience higher morbidity and mortality due to secondary organ injuries. The focus here is inflammatory arthritis with cardiac and lung complications. Current anti-arthritic drugs while may improve joint disease are not effective, or even detrimental, to the heart and lung. This unmet clinical need is more acute by lack of experimental models that recapitulate these clinical symptoms in order to investigate pathogenic mechanisms and test effective therapeutic approaches.

### Results

Arthritis was induced in mice exposed to high blood levels of homocysteine. Hyper-homocysteinemia arthritic animals develop diastolic dysfunction (the left ventricle does not dilate properly) with reduced lung compliance. These pathological features could be prevented by administration of an agonist at the formyl-peptide receptor (FPR) 2 through the reduction in cardiac fibroblasts (e.g. Pdpn$^+$Thy1.2$^+$ proinflammatory fibroblasts) and lung M1-like macrophages (together with a reduction in pro-fibrotic genes like nestin). In contrast, a dual FPR1/FPR2 agonist prevented the development of heart disease but failed to correct the lung dysfunction: this was due to an FPR1-mediated recruitment of inflammatory monocytes and concordantly increased numbers of pro-fibrotic CX$_3$CR1$^+$CD11c$^+$SiglecF$^+$MHCII$^{hi}$ macrophages. In human monocytes, the FPR1/2 agonist—but not the selective FPR2 agonist—synergised with CCL2 to promote cell chemotaxis and downstream CCR2-mediated signalling.

### Impact

Our data provide novel insights into the mechanisms associated with secondary organ injury in arthritis, revealing tissue-specific alterations in cell types and numbers. Moreover, they provide proof-of-concept for novel therapeutic strategies aiming at the control of secondary organ injury in arthritis. In the medium to long term, this study can be beneficial to mitigate, if not prevent, morbidity and mortality events experienced by RA patients.

## Macrophage/fibroblast co-cultures

Human macrophage cultures or co-cultures with fibroblasts, were conducted with monocyte-derived macrophages obtained after 7-day culture with 20 ng/ml hM-CSF. Cells were not checked for mycoplasma contamination. Cells were then polarised with LPS/interferon gamma or with IL-4 as published (McArthur et al, 2020), with or without BMS235 incubation. Some experiments were terminated after 24 h, and macrophage markers quantified by flow cytometry. In other cases, cells were added to six-well 3 µm pore Transwell™ inserts (Corning Cat No. 353091) onto a six-well plate of synovial fibroblasts or cardiac fibroblasts (purchased from Promocell, Dorset, UK). After 24 h in complete RPMI media, fibroblasts were harvested and analysed by flow cytometry or qPCR as above.

## Graphs

Some panels in the Figures as well as the synopsis were created with BioRender.com according to the licence of William Harvey Research Institute, Queen Mary University of London.

## Statistics

Data were analysed using GraphPad (Prism) v8 and v9, respectively. Two groups were compared using unpaired *t* test as appropriate. Brown-Forsythe test for equal variance was applied. Multiple groups were compared using a one-way ANOVA followed by Tukey's multiple comparisons test. Depending on sample distribution, a Welch's ANOVA test was performed followed by Tamhane T2 multiple comparisons test or a Kruskal–Wallis ANOVA followed by Dunn's multiple comparisons test. Multiparameter data with subsequent time points and multiple groups were compared using two-way ANOVA. A *P* value less than 0.05 was considered statistically significant.

# Data availability

Original files for flow cytometry and echocardiography are archived at http://www.ebi.ac.uk/biostudies/studies/S-BSST1861, whereas histology images are available at http://www.ebi.ac.uk/biostudies/bioimages/studies/S-BIAD1635.

The source data of this paper are collected in the following database record: biostudies:S-SCDT-10_1038-S44321-025-00227-1.

# Peer review information

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

## Acknowledgements

The authors would like to thank the Lead Discovery & Optimisation group at Bristol Myers Squibb for support with Luminex analyses; Dr. Shahid Tannu for help with tissue analyses; Prof. Marzia Malcangio (Kings College London) for helpful discussion on the study. The authors thank Dr. Elisa Corsiero (Queen Mary University of London) for the supply of rheumatoid arthritis synovial fibroblasts (anonymised samples). The authors acknowledge Drs. Loic Rolas and Mathieu-Benoit Voisin for guidance and reagents and the CMR Advanced Bio-Imaging Facility of QMUL for the microscopy analyses. This work is aligned with the British Heart Foundation Accelerator Award to Queen Mary, which focuses on cardiac inflammation. Moreover, it has been facilitated by the National Institute for Health Research Biomedical Research Centre at Barts Health NHS Trust. The study was funded by an unrestricted programme grant between Bristol Myers Squibb to Queen Mary University of London (MP and LVN). Senior fellowship of Versus Arthritis UK grant 22235 (LVN). Career development fellowship of Versus Arthritis UK grant 22855 (JC). Studentship of the Chernajovsky Foundation (MC). German Research Foundation (DFG) grants MA9604/1-1, grant number 460682455 and MA9604/3-1 CRC1450-C05, grant number 431460824 (AM). Multiphoton microscope was funded by a British Heart Foundation infrastructure grant (IG/17/2/32993) and other microscopes from Barts Charity (grant U0026) and QMUL infrastructure award.

## Author contributions

**Andreas Margraf**: Conceptualisation; Data curation; Formal analysis; Funding acquisition; Investigation; Visualisation; Methodology; Writing—original draft; Project administration; Writing—review and editing. **Jianmin Chen**: Conceptualisation; Data curation; Formal analysis; Funding acquisition; Investigation; Visualisation; Methodology; Project administration; Writing—review and editing. **Marilena Christoforou**: Data curation; Formal analysis; Investigation; Visualisation; Methodology. **Pol Claria-Ribas**: Data curation; Formal analysis; Investigation; Methodology. **Ayda Henriques Schneider**: Data curation; Formal analysis; Investigation; Methodology. **Chiara Cecconello**: Investigation; Methodology. **Weifeng Bu**: Data curation; Investigation;

Methodology. **Paul R C Imbert**: Investigation; Methodology; Writing—review and editing. **Thomas D Wright**: Data curation; Formal analysis; Investigation; Methodology. **Stefan Russo**: Data curation; Investigation; Methodology. **Isobel A Blacksell**: Investigation; Methodology. **Duco S Koenis**: Investigation; Methodology. **Jesmond Dalli**: Investigation; Methodology; Writing—review and editing. **John A Lupisella**: Investigation; Project administration; Writing—review and editing. **Nicholas R Wurtz**: Project administration; Writing—review and editing. **Ricardo A Garcia**: Investigation; Methodology; Project administration; Writing—review and editing. **Dianne Cooper**: Conceptualisation; Funding acquisition; Investigation; Methodology; Project administration; Writing—review and editing. **Lucy V Norling**: Conceptualisation; Data curation; Formal analysis; Supervision; Funding acquisition; Investigation; Methodology; Project administration; Writing—review and editing; contribution to writing. **Mauro Perretti**: Conceptualisation; Formal analysis; Supervision; Funding acquisition; Visualisation; Methodology; Writing—original draft; Project administration; Writing—review and editing.

Source data underlying figure panels in this paper may have individual authorship assigned. Where available, figure panel/source data authorship is listed in the following database record: biostudies:S-SCDT-10_1038-S44321-025-00227-1.

## Disclosure and competing interests statement

MP declares to be a shareholder of ResoTher Pharma ApS and a director of William Harvey Research Limited; advisory board member for SynAct Pharma AB; and is involved in the following commercial projects: SynAct Pharma AB and TXP Pharma AG. MP and JD are inventors on a patent related to AnxA1 pro-resolving peptides (European Patent 3533457 B1). JAL, NRW and RAG are employees of Bristol Myers Squibb and hold shares in the company.

# Expanded View Figures

## A

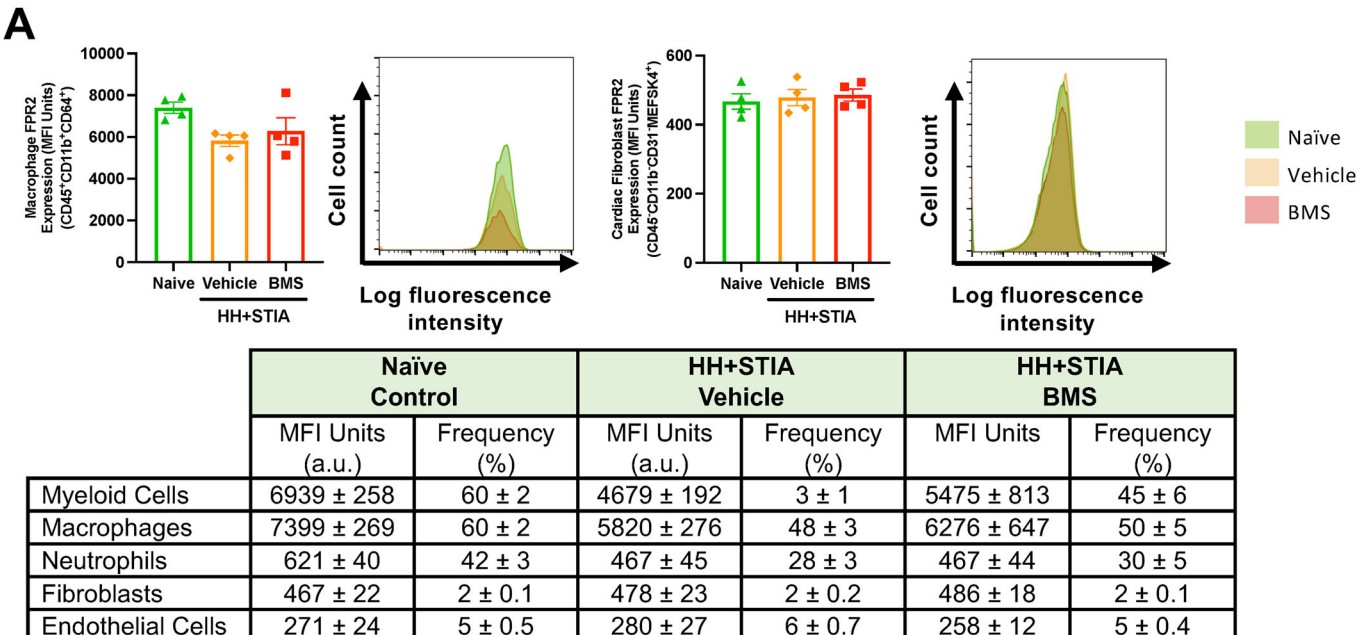

| | Naïve Control | | HH+STIA Vehicle | | HH+STIA BMS | |
|---|---|---|---|---|---|---|
| | MFI Units (a.u.) | Frequency (%) | MFI Units (a.u.) | Frequency (%) | MFI Units (a.u.) | Frequency (%) |
| Myeloid Cells | 6939 ± 258 | 60 ± 2 | 4679 ± 192 | 3 ± 1 | 5475 ± 813 | 45 ± 6 |
| Macrophages | 7399 ± 269 | 60 ± 2 | 5820 ± 276 | 48 ± 3 | 6276 ± 647 | 50 ± 5 |
| Neutrophils | 621 ± 40 | 42 ± 3 | 467 ± 45 | 28 ± 3 | 467 ± 44 | 30 ± 5 |
| Fibroblasts | 467 ± 22 | 2 ± 0.1 | 478 ± 23 | 2 ± 0.2 | 486 ± 18 | 2 ± 0.1 |
| Endothelial Cells | 271 ± 24 | 5 ± 0.5 | 280 ± 27 | 6 ± 0.7 | 258 ± 12 | 5 ± 0.4 |

## B

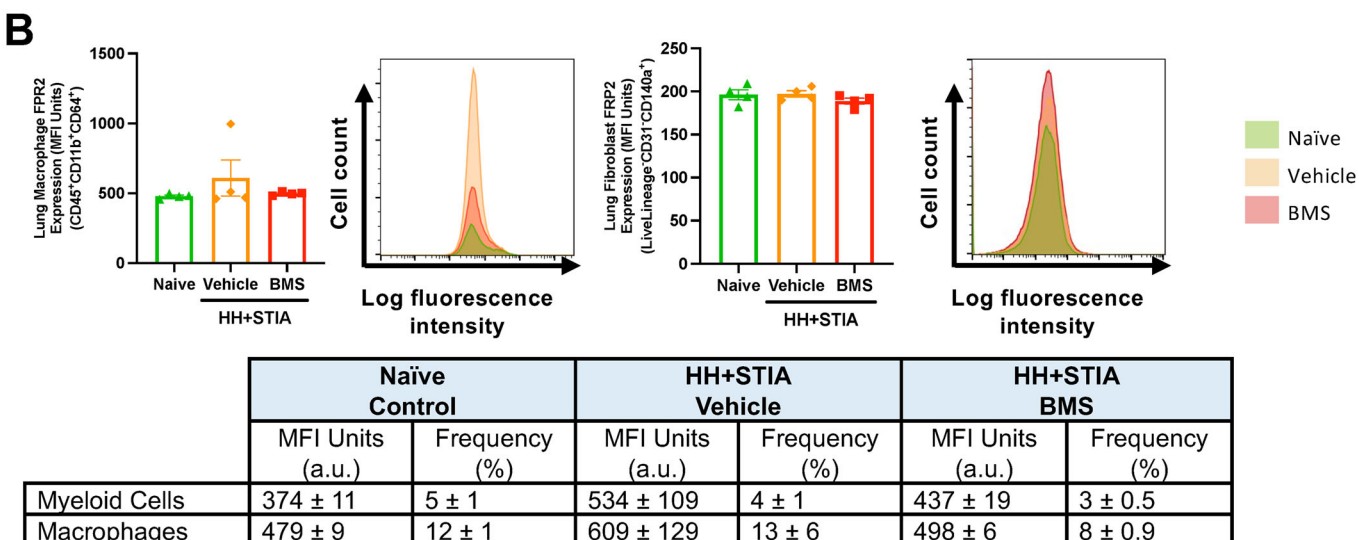

| | Naïve Control | | HH+STIA Vehicle | | HH+STIA BMS | |
|---|---|---|---|---|---|---|
| | MFI Units (a.u.) | Frequency (%) | MFI Units (a.u.) | Frequency (%) | MFI Units (a.u.) | Frequency (%) |
| Myeloid Cells | 374 ± 11 | 5 ± 1 | 534 ± 109 | 4 ± 1 | 437 ± 19 | 3 ± 0.5 |
| Macrophages | 479 ± 9 | 12 ± 1 | 609 ± 129 | 13 ± 6 | 498 ± 6 | 8 ± 0.9 |
| Neutrophils | 576 ± 11 | 48 ± 2 | 636 ± 90 | 54 ± 12 | 559 ± 36 | 45 ± 6 |
| Fibroblasts | 196 ± 6 | 2 ± 0.1 | 197 ± 4 | 2 ± 0.2 | 189 ± 3 | 2 ± 0.5 |
| Endothelial Cells | 189 ± 5 | 3 ± 0.1 | 173 ± 6 | 3 ± 0.3 | 166 ± 8 | 3 ± 0.6 |

Figure EV1. Formyl peptide receptor 2 (FPR2) expression in cardiac and lung cells.

HH + STIA was induced as in Fig. 1. From week 4, mice were treated with either vehicle (100 µl per os daily) or BMS235 (3 mg/kg per os daily). At the end of week 6, hearts and lungs were harvested, digested and processed for flow cytometry analysis. (A) Top panels, representative histograms for FPR2 expression on cardiac macrophages and fibroblasts. Table, summary data for the reported cardiac cell types. (B) Top panels, representative histograms for FPR2 expression on lung macrophages and fibroblasts. Table, summary data for the reported lung parenchyma cell types. Cardiac cell populations were defined by the following markers: myeloid cells, CD45+CD11b+; macrophages, CD45+CD11b+CD64+; neutrophils, CD45+CD11b+Ly6G+; endothelial cells, CD45-CD11b-CD31+; fibroblasts, CD45-CD11b-CD31-MEFSK4+. Lung cell populations were defined by the following markers: myeloid cells, CD45+CD11b+; macrophages, CD45+CD11b+CD64+; neutrophils, CD45+CD11b+Ly6G+; lung endothelial cells, Lineage-CD31+; fibroblasts, Lineage-CD31-CD140a+. Data are mean ± SEM of $n = 4$ (naive), $n = 4$ (HH + STIA+Vehicle) and $n = 4$ (HH + STIA + BMS) mice.

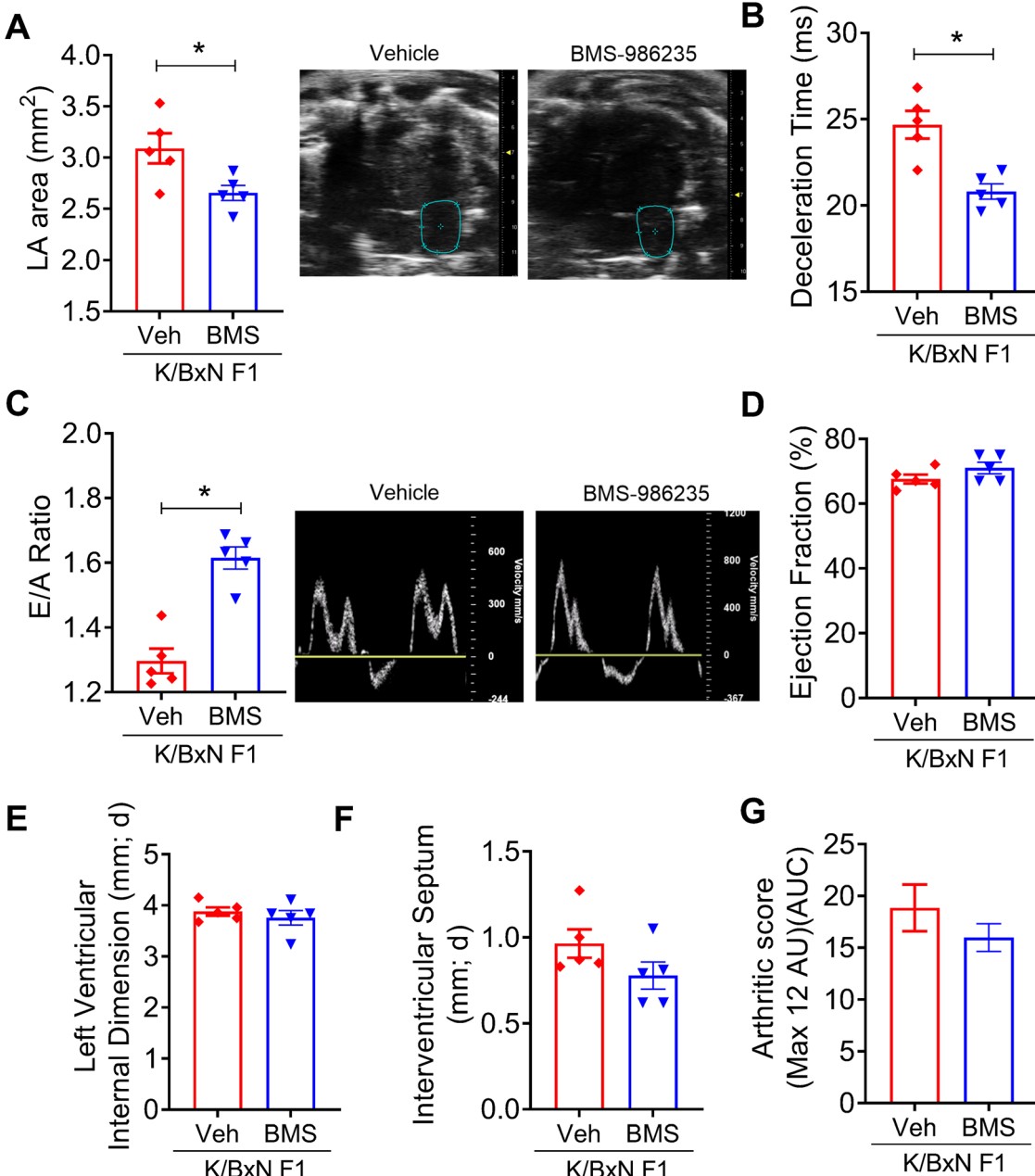

**Figure EV2. Selective FPR2 agonism by BMS235 significantly attenuates cardiac diastolic dysfunction in K/BxN F1 mice.**

K/BxN F1 mice were treated daily from week 4 to week 8 with vehicle or BMS235 (3 mg/kg p.o.). Echocardiography was performed at week 8 after 4 weeks of treatment. **(A)** Quantification of left atrial (LA) area (*, $P$ value = 0.0029); Right-hand images: representative B-mode four-chamber echocardiograms and left atrial (light blue circles) area in K/BxN F1 mice. **(B)** Quantification of deceleration time (*, $P$ value = 0.003). **(C)** Quantification of E/A ratio; Right-hand images: representative mitral flow patterns from pulsed-wave colour Doppler echocardiography (*, $P$ value = 0.0003). **(D–G)** Quantification of four other cardiac parameters in the two experimental groups; **(D)** ejection fraction; **(E)** left ventricular dimension in diastolic (d) phase; **(F)** interventricular septum in diastolic (d) phase; **(G)** arthritic score, area under curve (AUC). Data are mean ± SEM of $n = 5$ mice per group. *$P < 0.05$ vs. vehicle group. One-way ANOVA with Tukey's multiple comparisons test. For (**B**, **J**): Kruskal–Wallis ANOVA followed by Dunn's multiple comparisons test.

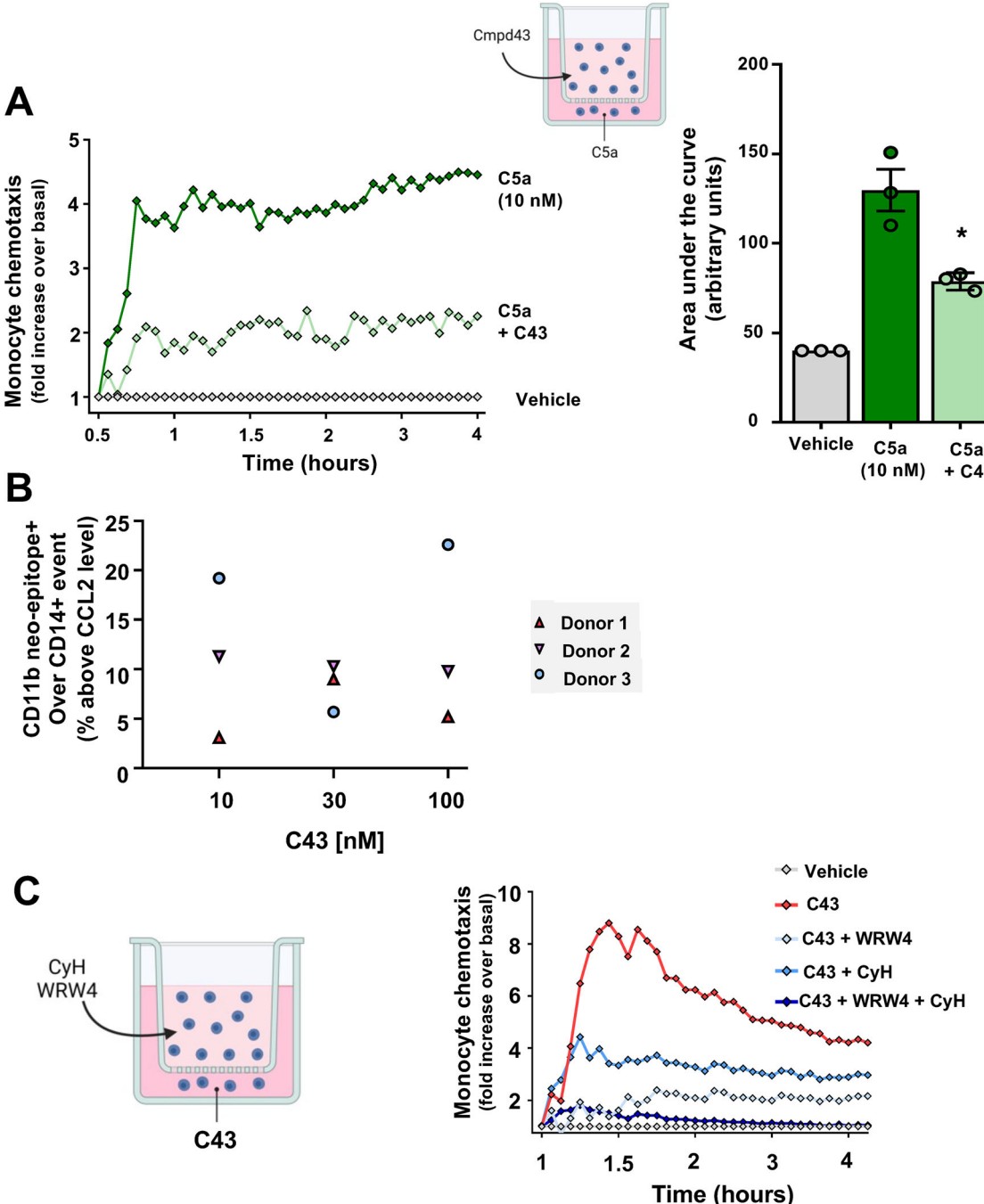

**Figure EV3. Human monocyte reactivity experiments.**

Chemotaxis of purified human peripheral blood monocytes was assessed using a xCELLigence™ DP system as in Fig. 8. (**A**) Inhibition of C5a (10 nM) induced monocyte chemotaxis by C43 (100 nM; 30 min pre-incubation); *P value = 0.0051 vs. C5a data. (**B**) Additive effect of C43 to CCL2 (10 nM)-mediated neo-epitope expression. Data for each single donor (n = 3 in total) are presented. (**C**) C43 (100 nM)-mediated human monocyte chemotaxis; regulation by cyclosporin H (CyH, 1 μM) and WRW4 (1 μM) added to cells 30 min prior to beginning of the chemotaxis assay using the xCELLigence™ DP system. Quantitative data from multiple donor cells are given in the main text. For AUC-values: One-way ANOVA with Tukey's multiple comparisons test.

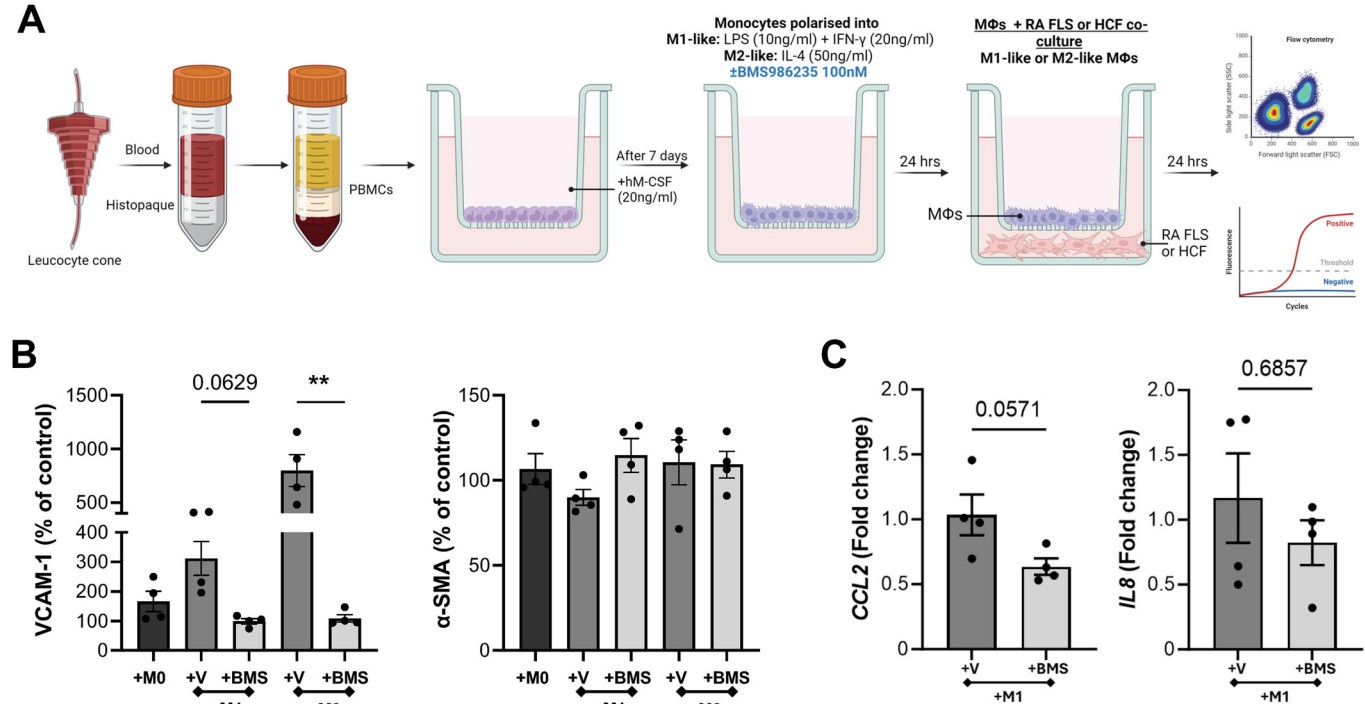

**Figure EV4. Modulation of human macrophage/fibroblast crosstalk by BMS235.**

(**A**) Schematic of the co-culture experiments with human monocyte-derived macrophages treated with vehicle or BMS235 (BMS, 100 nM) or vehicle control (0.1% DMSO) for 24 h prior to addition to either human RA synovial fibroblasts (FLS) or human cardiac fibroblasts (HCF). Fibroblast markers were quantified 24 h later. (**B**) VCAM-1 and α-SMA expression in human RA fibroblast-like synoviocytes (FLS). Data are mean ± SEM of 3–4 distinct cone preparations. (**, adjusted *P* value = 0.0025), one-way nonparametric Kruskal–Wallis, Dunn's multiple comparisons test. (**C**) Chemokine gene expression in human cardiac fibroblasts (HCF), where Ct values were normalised using 18S as a housekeeping gene and fold change was calculated relative to the geometric mean of the HCF co-cultured with vehicle control M1-like macrophages. Data are mean ± SEM of 4 distinct macrophage preparations. Statistical analysis is nonparametric *t*-test Mann-Whitney test.

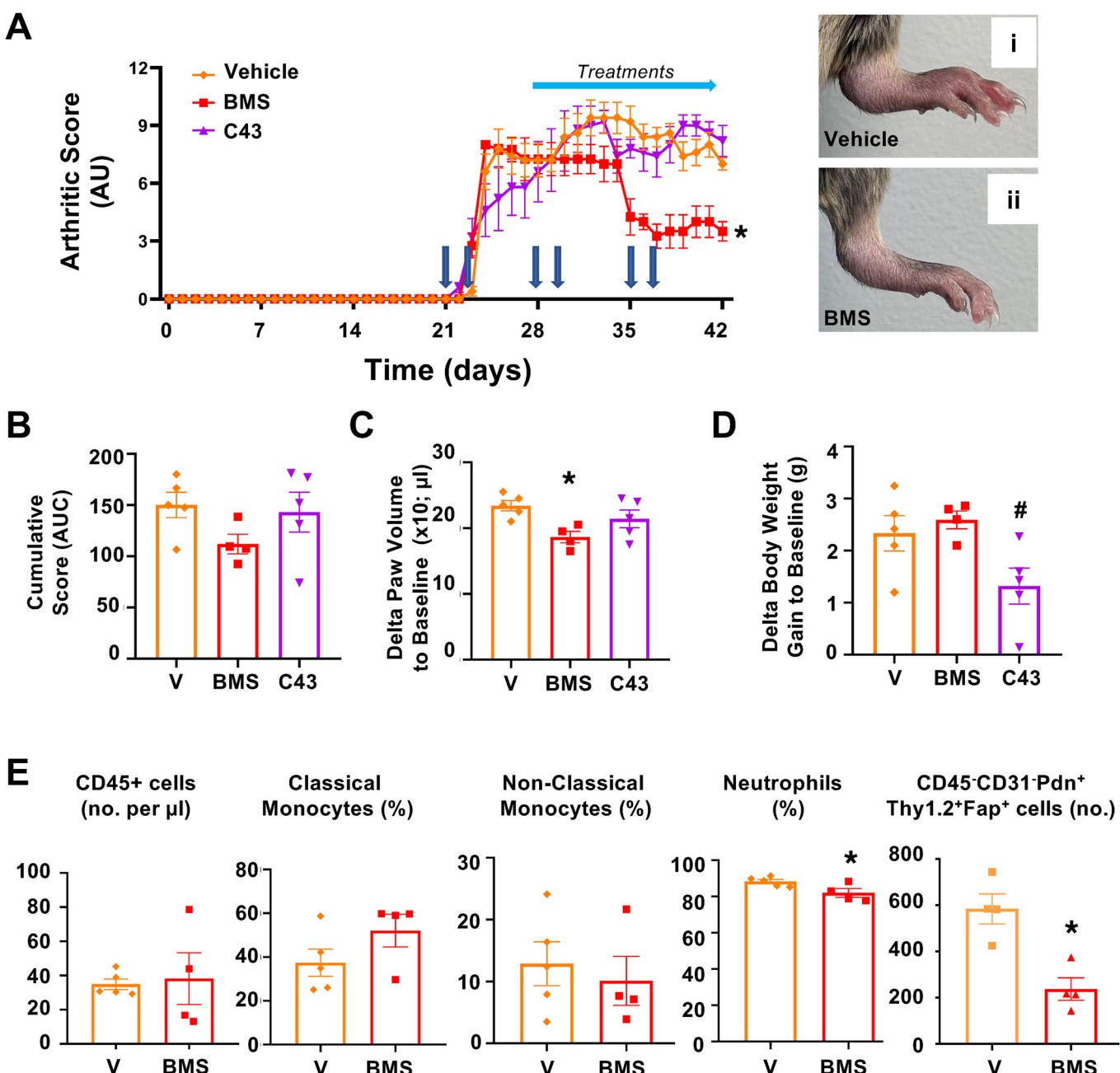

**Figure EV5. Impact of FPR2 agonism on joint disease.**

Figure EV5. Impact of FPR2 agonism on joint disease. HH + STIA was obtained as in Fig. 1. From week 4, mice were treated with either vehicle or BMS235 (3 mg/kg per os) or C43 (10 mg/kg per os) daily. (A) Time course of the arthritic score (arrows indicate serum injections; bar, treatment time). Representative images of arthritic score: i) score ≥9, ii) score ~6. (B) Cumulative value for the arthritic score. Area under the curve (AUC). (C) Oedema shown as delta paw volume between day 0 and day 42 (*, adjusted $P$ value = 0.0085). (D) Change in body weight between day 0 and day 42 (#, adjusted $P$ value = 0.0155). (E) Cellular characterisation from paws collected at day 42. Classical monocytes: CD45+CD11b+CD115+CD43-Ly6Chigh; non-classical monocytes: CD45+CD11b+CD115+CD43+Ly6Clow. Neutrophils: CD45+CD11b+CD115-Ly6G+ ($P$ value = 0.0398). Proinflammatory fibroblasts: Thy1.2+CD45-CD31-Pdpn+Fap+ ($P$ value = 0.0053). Data are mean ± SEM of $n$ = 4–5 mice per group. *$P < 0.05$ vs. vehicle; #$P < 0.05$ vs. BMS group. For (A): Two-way ANOVA with Tukey's multiple comparisons test. For (B–D): one-way ANOVA with Tukey's multiple comparisons test. For (E): unpaired two-tailed $t$-test.

