## [Peer Review File · EMBO Molecular Medicine]

Formyl-peptide receptor type 2 activation mitigates heart and lung damage in inflammatory arthritis

Andreas Margraf, Jianmin Chen, Marilena Christoforou, Pol Claria-Ribas, Ayda Henriques Schneider, Chiara Ceconello, Weifeng Bu, Paul Imbert, Thomas Wright, Stefan Russo, Isobel Blacksell, Duco Koenis, Jesmond Dalli, John Lupisella, Nicholas Wurz, Ricardo Garcia, Dianne Cooper, Lucy Norling, and Mauro Perretti

Corresponding authors: Mauro Perretti (m.perretti@qmul.ac.uk) , Lucy Norling (l.v.norling@qmul.ac.uk)

Review Timeline:

Submission Date:	1st Aug 24
Editorial Decision:	3rd Sep 24
Revision Received:	10th Feb 25
Editorial Decision:	25th Feb 25
Revision Received:	12th Mar 25
Accepted:	17th Mar 25

Editor: Lise Roth

Transaction Report:

3rd Sep 2024

Dear Prof. Perretti,

Thank you for the submission of your manuscript to EMBO Molecular Medicine, and please accept my apologies for the delay in getting back to you in this busy time of the year. We have now received feedback from the three reviewers who agreed to evaluate your manuscript. As you will see from the reports below, the reviewers all mention the potential interest of the findings, but also raise several partially overlapping concerns including the unclear rationale to focus on FPR2, the lack of spatial cell analysis, and the limited mechanistic insight.

If you feel you can satisfactorily address the referees' concerns, you may wish to submit a revised version of your manuscript. As mentioned by referee #2, only a selected number of experiments could be repeated using the model from J. Hill's lab. Please attach a covering letter giving details of the way in which you have handled each of the points raised by the referees. Acceptance of the manuscript will entail a second round of review. EMBO Molecular Medicine encourages a single round of revision only and therefore, acceptance or rejection of the manuscript will depend on the completeness of your responses included in the next, final version of the manuscript. For this reason, and to save you from any frustrations in the end, I would strongly advise against returning an incomplete revision.

We are expecting your revised manuscript within three months, if you anticipate any delay, please contact us.

We require:

4) A .docx formatted letter INCLUDING the reviewers' reports and your detailed point-by-point responses to their comments. As part of the EMBO Press transparent editorial process, the point-by-point response is part of the Review Process File (RPF), which will be published alongside your paper.

5) A complete author checklist, which you can download from our author guidelines (<https://www.embopress.org/page/journal/17574684/authorguide#submissionofrevisions>). Please insert information in the checklist that is also reflected in the manuscript. The completed author checklist will also be part of the RPF.

6) All Materials and Methods need to be described in the main text using our 'Structured Methods' format, which is required for all research articles. According to this format, the Methods section includes a Reagents and Tools Table (listing key reagents, experimental models, software and relevant equipment and including their sources and relevant identifiers) followed by a Methods and Protocols section describing the methods using a step-by-step protocol format. The aim is to facilitate adoption of the methodologies across labs. More information on how to adhere to this format as well as a downloadable template (.docx) for the Reagents and Tools Table can be found in our author guidelines: <https://www.embopress.org/page/journal/17574684/authorguide#structuredmethods>.

7) Please note that all corresponding authors are required to supply an ORCID ID for their name upon submission of a revised manuscript.

8) It is mandatory to include a 'Data Availability' section after the Materials and Methods. Before submitting your revision, primary datasets produced in this study need to be deposited in an appropriate public database, and the accession numbers and database listed under 'Data Availability'. Please remember to provide a reviewer password if the datasets are not yet public (see <https://www.embopress.org/page/journal/17574684/authorguide#dataavailability>).

9) For data quantification: please specify the name of the statistical test used to generate error bars and P values, the number (n) of independent experiments (specify technical or biological replicates) underlying each data point and the test used to calculate p-values in each figure legend. The figure legends should contain a basic description of n, P and the test applied. Graphs must include a description of the bars and the error bars (s.d., s.e.m.). Please provide exact p values.

10) Our journal encourages inclusion of *data citations in the reference list* to directly cite datasets that were re-used and obtained from public databases. Data citations in the article text are distinct from normal bibliographical citations and should directly link to the database records from which the data can be accessed. In the main text, data citations are formatted as follows: "Data ref: Smith et al, 2001" or "Data ref: NCBI Sequence Read Archive PRJNA342805, 2017". In the Reference list, data citations must be labeled with "[DATASET]". A data reference must provide the database name, accession number/identifiers and a resolvable link to the landing page from which the data can be accessed at the end of the reference. Further instructions are available at .

11) We replaced Supplementary Information with Expanded View (EV) Figures and Tables that are collapsible/expandable online. A maximum of 5 EV Figures can be typeset. EV Figures should be cited as 'Figure EV1, Figure EV2" etc... in the text and their respective legends should be included in the main text after the legends of regular figures.

12) Author contributions: CRediT has replaced the traditional author contributions section because it offers a systematic machine readable author contributions format that allows for more effective research assessment. Please remove the Authors Contributions from the manuscript and use the free text boxes beneath each contributing author's name in our system to add specific details on the author's contribution. More information is available in our guide to authors.

13) Disclosure statement and competing interests: We updated our journal's competing interests policy in January 2022 and request authors to consider both actual and perceived competing interests. Please review the policy <https://www.embopress.org/competing-interests> and update your competing interests if necessary.

14) Every published paper now includes a 'Synopsis' to further enhance discoverability. Synopses are displayed on the journal webpage and are freely accessible to all readers. They include a short stand first (maximum of 300 characters, including space) as well as 2-5 one-sentences bullet points that summarizes the paper. Please write the bullet points to summarize the key NEW findings. They should be designed to be complementary to the abstract - i.e. not repeat the same text. We encourage inclusion of key acronyms and quantitative information (maximum of 30 words / bullet point). Please use the passive voice. Please attach these in a separate file or send them by email, we will incorporate them accordingly.

Please also suggest a striking image or visual abstract to illustrate your article as a PNG file 550 px wide x 300-600 px high. A cropped portion of this image will serve as thumbnail for the table of content on our webpage.

15) As part of the EMBO Publications transparent editorial process initiative (see our Editorial at <http://embomolmed.embopress.org/content/2/9/329>), EMBO Molecular Medicine will publish online a Review Process File (RPF) to accompany accepted manuscripts.

In the event of acceptance, this file will be published in conjunction with your paper and will include the anonymous referee reports, your point-by-point response and all pertinent correspondence relating to the manuscript. Let us know whether you agree with the publication of the RPF and as here, if you want to remove or not any figures from it prior to publication. Please note that the Authors checklist will be published at the end of the RPF.

I look forward to receiving your revised manuscript.

Yours sincerely,

Lise Roth

***** Reviewer's comments *****

Referee #1 (Remarks for Author):

Review -

Title - Formyl-peptide receptor type 2 activation mitigates secondary organ damage in inflammatory arthritis

Summary -

Targeting the systemic inflammatory effects of RA - and other disorders - that lead to cardiovascular and pulmonary dysfunction is a very important and often ignored area needing further investigation and also treatments

This work incorporates extensive study involving development of a new model of RA induced lung and heart inflammation and dysfunction - using a homocysteine model together with arthritogenic sera

This manuscript then provides an analysis of a potential target for new treatments

There is extensive work in these studies - a complex series of studies with multiple layers of experimental analyses

The concept and the targeted area of research are excellent - Some further explanation of the methods used, in particular the method used to produce the HH STIA mice, the numbers of experimental animals assessed and who performed the blinded analyses would strengthen the paper. The numbers of mice in each experimental group appear small but significance is achieved in some of the analyses. Reference to prior work with HH and/ or STIA induced arthritis models would be helpful, specifically defining the source of the arthritic serum. What is the proposed/ postulated and / or the known mechanism of action of the formyl peptide receptor in improving systemic responses to arthritis - a diagram would be helpful

Questions -

Is the homocysteine model completely new - "Since RA patients exhibit metabolic alterations including elevated plasma levels of homocysteine (Roubenoff et al., 1997; Schroecksadel et al., 2003), we hypothesized a functional link between hyper-homocysteinemia and lung or heart dysfunction in arthritis. To investigate this, we employed a model of STIA in hyper-homocysteine mice and..."

Define arthritogenic serum - not defined in methods - what is the source, composition and concentration

Are neutrophils typically increased in systemic RA CV and pulmonary responses?

How many mice were used in initial model development - this is not provided in the methods or text but are provided in Figure 2 Legend as follows "Data are mean{plus minus}SEM of n=3 (naïve), n=4 (HH) and n=5 (STIA and HH+STIA) mice." Were any mice lost to side effects in the models. Was a power calculation performed to assess whether these numbers of mice are acceptable to achieve significance? However, In Figure 5 with Fpr agonism there are 5 mice per treatment group and significance is reported

The numbers of mice vary with each experiment - n = 3 to n = 5 - this should be explained

A table provided numbers of mice in each experimental group and the analyses performed on each mouse group would be helpful. Were all analyses performed on all mice?

In figure 3 the M2 Macrophage and the profibrotic macrophage seem significantly increased for HH but not STIA + HH - on visual inspection - what were the p values?

Why homocysteine - a proposed mechanism is resented - This is a marker for disease.

What other potential effects other than reducing systemic inflammation - what other targets might be proposed for the formyl peptide receptor

Who read the echo's to determine the cardiac dysfunction - a blinded read of echo and histology would be best - Was there reduce EF in any of the mice. Was vascular disease assessed as a source for inflammatory disease sustemically - this likely would be part of future experiments

Can the authors comment on the potential for Fpr agonism based on human data analyses to be similar in mice and human subjects

Is there a positive control - ? NSAID ASA or steroids

Any adverse effects?

"Fpr2 null mice develop early onset diastolic dysfunction, which is associated with reduced survival upon ageing due to a prolonged failure of resolution responses (Tourki et al, 2020). This study indicates that Fpr2 exerts a tonic protective role against diastolic dysfunction. " Testing an activator of fpr2 is logical

Minor comments-

Figure 8 - the symbols upper left in the top left panel do not match the symbols in the figure

The IL23R gene expression changes also look to be of interest -just a comment

Lung compliance measurement seems appropriate

Flow cytometry is subjective albeit generally an accepted and an excellent tool - were the organs weighed

Were histology samples taken for analysis of structural changes in heart and lung and arthritis?

Define abbreviations such as E/A ratio on echo when first used

Referee #2 (Remarks for Author):

The manuscript by Margraf et al. provides an interesting approach for examining potential mechanisms for systemic complications of rheumatoid arthritis. This is a critical topic and one highly worthy of investigation. The manuscript is nicely written. To this reviewer, the approach has the added (potential) advantage of being able to examine complications in two organ systems. Is this potential advantage being acted up? This reviewer will highlight a few areas/suggestions and this will be followed by a series of questions clarification.

There are many strengths to this excellent manuscript. It is process/data of the selection of the Fpr2 target that requires clarification. In the results section the data in support of the target selection moves from human to rodent, from experimental to publicly available, all with differing degrees of "validation. Filling some of the gaps would be helpful. The section would benefit from highlighting/adding in "validations." This reviewer is not confident in the logic that because Fpr2 KO mice have diastolic heart failure --- that agonist tone should "maintaining" normal diastolic function. Understanding that there are more unknowns than knows, if there is "basal" activation and based on the data presented the receptor is up regulated, in the absence of loss of ligand then activity should be higher; not lower. The process of the selection is directly relevant regardless of the "outcome" of the target selection process; which appears to have been correct based on the therapeutic study. Understanding that there are more unknowns than knows, an argument could be made that if there is "basal" activation and based on the data presented the receptor is up regulated, in the absence of loss of ligand then activity should actually be higher; not lower. Detail on the target, its degree of activation, and cell specificity of expression/activation could go a long way to bridging that gap.

The second area that requires attention is the homocysteine administration for heart failure with preserved ejection fraction model. There have been long-standing questions on the value/applicability of any mouse models in this area. The general consensus now would be that the model from Dr. Joseph Hill's lab (UTSW), has gained traction (for today) as the best model. Asking the authors to reproduce all of the data in this more established heart failure model is not reasonable. Perhaps selected parts could be repeated. The citation and another recent paper are:

Nitrosative stress drives heart failure with preserved ejection fraction - PubMed (nih.gov)

Mouse Model of Heart Failure With Preserved Ejection Fraction Driven by Hyperlipidemia and Enhanced Cardiac Low-Density Lipoprotein Receptor Expression | Journal of the American Heart Association (ahajournals.org).

Minor areas for clarification:

1. It would be valuable know normal homocysteine levels and levels of the mice in this study, perhaps against human values. Information can be found that 10-12 micromol/L is the value for normal/control human subjects and 400 or greater for patients with homozygous abnormalities.

2. Data in the paper looks at the distribution of cells in heart/lung based on the use of serum injection with homocysteine in diet. The analysis by flow has the advantage of being quantitative but misses the opportunity to understand the spatial distribution of the cells, which could be useful. Also, this could also overcome the limitation on the success of accessing cells (i.e. macrophages) out of the tissue. Also, percents do not relate to total numbers which could be obtained with this "low tech" approach.
3. Studies were sex matched but were both sexes studied.
4. Supplemental table 2 -- Adiponectin 650000 +/- 0? Correct.
5. The sample size Stats on cytokines might need to be reviewed.

Referee #3 (Comments on Novelty/Model System for Author):

The authors previously found that K/BxN F1 mice developed HFpEF with the infiltration of activated T cells and the expansion of specific fibroblast subsets and that recombinant annexin A, an endogenous ligand of FPR2, ameliorated HFpEF. In this study, the authors established the murine model of RA complicated with HFpEF and reduced lung compliance using K/BxN serum transfer (STIA) and dietary supplementation of homocysteine. Comparing these two similar models, the former seems to be T-cell related, the latter macrophage-related. Thus this model would be novel but not established yet.

Referee #3 (Remarks for Author):

In this study, the authors established the murine model of rheumatoid arthritis complicated with both cardiac and pulmonary dysfunctions by the combination of K/BxN serum transfer (STIA) and dietary supplementation of homocysteine (HH). The cardiac dysfunction induced in this STIA+HH model resembled HFpEF, frequently found in RA patients, histologically characterized by the infiltration of neutrophils and the expansion of specific fibroblast subsets, while the pulmonary dysfunction with reduced compliance accompanied by the increased M1/M2 macrophage ratio in the interstitial tissue. By analyzing public RNA-seq data from RA and HC PBMCs, the authors focused on FPR2, a GPCR for the pro-resolving mediators such as resolvin D1 as a therapeutic target. The authors examined a FPR2-selective agonist, BMS235, and a dual FPR1/2 agonist, C43. BMS235 ameliorated arthritis, and cardiac and pulmonary dysfunction, but C43 did not, except for the cardiac dysfunction. The authors demonstrated that C43 failed to improve lung compliance because of its FPR1-mediated potentiation of CCL2-mediated monocyte chemotaxis.

The manuscript is well-described. However, several points need to be clarified before the manuscript can be considered for publication.

#1. The authors previously found that K/BxN F1 mice developed HFpEF with the infiltration of activated T cells and the expansion of specific fibroblast subsets and that recombinant annexin A, an endogenous ligand of FPR2, ameliorated HFpEF. In this study, the authors established the murine model of RA complicated with HFpEF and reduced lung compliance. This inducible model by K/BxN serum transfer (STIA) and dietary supplementation of homocysteine (HH) may be helpful in investigating the specific gene function using KO mice with a B6 background. However, several concerns should be addressed: 1) HFpEF in the STIA+HH model was histologically characterized by the infiltration of neutrophils, which was different from the infiltration of activated T cells found in K/BxN F1 mice. Which HFpEF resembles that often found in RA patients? 2) The authors demonstrated reduced lung compliance with the increased M1/M2 macrophage ratio in the interstitium assessed by flow cytometry. In addition to these analyses, histological and/or imaging analysis by micro-CT should be performed to evaluate the interstitial pneumonia pattern found in the STIA+HH model.

#2. The authors were interested in FPR2 by DEG analysis using public RNA-seq data from RA and HC PBMCs. However, FPR2 is a well-known receptor of pro-resolving mediators, and many studies, including the author's preceding study, demonstrated its anti-inflammatory and cardioprotective roles. Why did you focus on FPR2? Are there any rationales to select FPR2 from the unbiased DEG analysis?

#3. The reduced lung compliance in the STIA+HH model was alleviated by BMS235 but not C43. The authors found elevated numbers of Ly6ChighCCR2+ classical monocytes in the C43-treated lung and demonstrated that C43 potentiated the CCL2-mediated monocyte chemotaxis. This mechanism may explain why C43 had no effects on lung compliance even though both BMS235 and C43 had a similar effect on the M1/M2 macrophage ratio. However, the molecular mechanisms responsible for the effectiveness of BMS235 are more important and should be addressed in detail both in vivo and in vitro.

#4. The authors demonstrated that BMS235 but not C43 attenuated arthritis. The data is inconsistent with previous data reported by Kao et al., 2014. The reasons for this discrepancy should be discussed.

Referee #1

Targeting the systemic inflammatory effects of RA - and other disorders - that lead to cardiovascular and pulmonary dysfunction is a very important and often ignored area needing further investigation and also treatments. This work incorporates extensive study involving development of a new model of RA induced lung and heart inflammation and dysfunction - using a homocysteine model together with arthritogenic sera. This manuscript then provides an analysis of a potential target for new treatments. There is extensive work in these studies - a complex series of studies with multiple layers of experimental analyses.

The concept and the targeted area of research are excellent - Some further explanation of the methods used, in particular the method used to produce the HH STIA mice, the numbers of experimental animals assessed and who performed the blinded analyses would strengthen the paper. The numbers of mice in each experimental group appear small but significance is achieved in some of the analyses. Reference to prior work with HH and/ or STIA induced arthritis models would be helpful, specifically defining the source of the arthritic serum. What is the proposed/ postulated and / or the known mechanism of action of the formyl peptide receptor in improving systemic responses to arthritis - a diagram would be helpful

We thank the Reviewer for raising these concerns which we will address in detail below. Overall, application of 3R principles leads to a reduction in animals used per experiments. We relied on power calculations and pre-experiments in which we established this model.

We now included a graphical abstract that summarises mechanisms and actions of FPRs in this model arthritis with comorbidities. We furthermore highlighted a recent review of our group (Margraf and Perretti, 2022) in which we briefly address immunomodulatory features of FPRs.

Questions -

Q1. Is the homocysteine model completely new - "Since RA patients exhibit metabolic alterations including elevated plasma levels of homocysteine (Roubenoff et al., 1997; Schroecksadel et al., 2003), we hypothesized a functional link between hyper-homocysteinemia and lung or heart dysfunction in arthritis. To investigate this, we employed a model of STIA in hyper-homocysteine mice and..."

The protocol to increase circulating homocysteine levels is not new but based on previously published studies. However, a combination of this model with the of serum transfer induced arthritis (STIA) has never been reported before. This is why, in the present manuscript, we describe a 'model arm' and a 'pharmacological arm', as we needed to perform a degree of characterization of the HH+STIA model. Once secondary organ dysfunction was observed, we were ready for the follow up investigations.

Q2. Define arthritogenic serum - not defined in methods - what is the source, composition and concentration

The serum was prepared from the KBxN F1 progeny, obtained by crossing the KRN and NOD mouse colonies. We have expanded the sentence on page 21 that now reads "...This mouse colony was originally described by Mathis and Benoist (Kouskoff et al, 1996; Matsumoto et al, 1999). Animals were bred in-house, develop arthritis from 4 to 5 week of age and are routinely culled at week 14-15 to prepare the serum which is stored at -80°C prior to further use."

For further info, we batch test the serum for arthritis potency: in this experiment 100µl was given at the indicated time points to sustain arthritis (as shown in Figure 1).

Q3. Are neutrophils typically increased in systemic RA CV and pulmonary responses?

Neutrophils are key immune cells with an appreciated pathogenic role in joint disease (see review PMID: 30709614) both in man and mouse. In newly diagnosed RA patients, the neutrophil-to-lymphocyte ratio predicts treatment failure (PMID: 31248587). Similarly, high

neutrophil numbers are reported for both interstitial lung diseases (PMID: 37307262) as well as cardiovascular disease (PMID: 37950632), in all cases associated with worsened outcome.

Q4. How many mice were used in initial model development - this is not provided in the methods or text but are provided in Figure 2 Legend as follows "Data are mean{plus minus}SEM of n=3 (naïve), n=4 (HH) and n=5 (STIA and HH+STIA) mice." Were any mice lost to side effects in the models. Was a power calculation performed to assess whether these numbers of mice are acceptable to achieve significance? However, In Figure 5 with Fpr agonism there are 5 mice per treatment group and significance is reported. The numbers of mice vary with each experiment - n = 3 to n = 5 - this should be explained. We ran two sets of experiments, in essence. One set used N=3 to 5 mice to determine whether HH+STIA group developed signs of heart and lung dysfunction. Once this was completed (model arm of the study), we then ran a second experiment where N=5 mice were used for all groups (pharmacological arm of the study). We have reported statistical significance, using the indicated statistical analysis.

In our previous study with the KBxN F1 model (Chen et al. 2021) we used n=4 to n=6 mice per group, reaching statistical significance. In accordance with 3R principles, we have not scaled the number of animals used in each experimental group.

More specifically, data in current Fig. EV5, were generated with experimental groups formed by 5 mice each. However, within the BMS235-treated group, one mouse did not develop arthritis and was identified as an outlier during the arthritic score area under the curve (AUC) analysis using Grubbs' test ($\alpha=0.1$). Panel on the right, from the original submission, highlights this outlier.

We have now excluded this mouse from the whole of the data produced in this group (whereas in the original submission we removed it from a few analyses only, as correctly spotted by the Reviewer – thank you for this). Additionally, in the joint fibroblast graph in Fig. EV5 (right bar graph in Panel E), one mouse from the Vehicle group was removed as an outlier identified using the ROUT method (Q=1%).

Q5. A table provided numbers of mice in each experimental group and the analyses performed on each mouse group would be helpful. Were all analyses performed on all mice?

We have reported the number of mice and the statistical tests in each figure legend. Individual data points are also indicated to ensure data transparency

Q6. In figure 3 the M2 Macrophage and the profibrotic macrophage seem significantly increased for HH but not STIA + HH - on visual inspection - what were the p values?

We agree the data spread suggests a difference: calculated p values are presented in the Table below (statistical analyses: One way ANOVA with Tukey's test). We have added this information on Figure 3 panel D with the symbol §.

M2 Macrophages	Profibrotic macrophages	current App Fig. S2
STIA vs. HH+STIA p= 0.9921	STIA vs. HH+STIA p= 0.7858	STIA vs. HH+STIA p= 0.6488
STIA vs. HH p= 0.0010	STIA vs. HH p= 0.0999	STIA vs. HH p= 0.1695
STIA vs. naïve p=0.9772	STIA vs. naïve p=0.9298	STIA vs. naïve p=0.9573
HH+STIA vs. HH p=0.0006	HH+STIA vs. HH p=0.0207	HH+STIA vs. HH p=0.6928
HH+STIA vs. naïve p=0.9165	HH+STIA vs. naïve p=0.5135	HH+STIA vs. naïve p=0.9524
HH vs. naïve p=0.0060	HH vs. naïve p=0.3845	HH vs. naïve p=0.4821

Q7. Why homocysteine - a proposed mechanism is presented - This is a marker for disease.

Indeed, homocysteine is a marker for disease progression in a few pathologies. As an example, hyper-homocysteinemia has been linked to altered cardiac metabolism through generation of superoxide (PMID: 17200441). Moreover, it can enhance neutrophil-endothelial interactions through extracellular H₂O₂ (PMID: 10066675). In our context these pro-oxidant properties of homocysteine may be relevant, as well as its ability to promote fibroblast migration and activation (PMID: 36454114). Nonetheless, in the manuscript we had indicated a few potential modes of action (Page 18/19), stating also that future studies may address the specific cellular target of homocysteine in settings of arthritis.

Q8. What other potential effects other than reducing systemic inflammation - what other targets might be proposed for the formyl peptide receptor.

We propose that FPR2 activation modulates immune and stromal cell numbers and activity: these changes in cell target and phenotype are tissue specific. As such there may be a contribution of modulation of the degree of systemic inflammation, however it remains more plausible that tissue-specific alterations in cell phenotypes could underpin the functional effects of BMS235 and C43. In fact, we could not observe major changes in circulating factors and mediators. In addition, C43 but not BMS235 impacts on lung monocytes and pro-fibrotic macrophages.

The new data produced with the KBxN F1 colony (Figure EV2) corroborates this view, as BMS235 prevents cardiac dysfunction while being totally inactive on markers of joint disease.

Q9. Who read the echo's to determine the cardiac dysfunction - a blinded read of echo and histology would be best - Was there reduce EF in any of the mice. Was vascular disease assessed as a source for inflammatory disease systemically - this likely would be part of future experiments.

Echocardiography was evaluated by an experienced medical doctor trained in mouse echocardiography. Analyses were performed blinded. Ejection Fraction was not reduced in any of the groups, as already reported in the original Figure 2B. Aside from echocardiography, all analyses were conducted in a blinded fashion. For each analysis, operators were not aware of the experimental groups.

Yes indeed, future work around the pathogenic properties of hyper-homocysteine will focus on the vasculature and potential changes that may occur there (e.g. NO and H₂S release; degree of calcification of smooth muscle cells). As Reviewer #1 indicates, this will be part of future studies.

Q10. Can the authors comment on the potential for Fpr agonism based on human data analyses to be similar in mice and human subjects

We have been working on the pharmacology of Formyl-peptide receptors for some time. In our hands, and published data from us and other groups, FPR2 agonists hold great potential for therapeutic application. As indicated in the manuscript, BMS235 has completed Phase 1 studies. Another FPR2 agonist, peptide RTP-026 has been tested in satisfactory Phase 1 trial and just entered Phase 2a trial in heart attack patients. All in all, we believe that a good parallel can be drawn between mouse and human efficacy for FPR2 agonists.

More specifically, preclinical studies with rodent cells or in-vivo rodent systems have been complemented and confirmed in studies with human cells (see for example ref 16, PMID: 34466754). In addition, our work on mouse macrophages (e.g. see PMID: 20107188) has been confirmed and complemented with human macrophages (PMID: 20107188).

Moreover, on top of our scientific reasoning presented on Page 7 (FPR2 in RA immune cells and the phenotype of Fpr2 KO mice), we now cite a study which reports FPR2 expression in human heart, further substantiating target selection (PMID: 36528331) (new Bouhadoun A et

al., 2023, in revised manuscript). In addition, we have run a new set of experiments to quantify Fpr2 expression in cardiac and lung cells from HH+STIA mice, with or without BMS235 treatment. The data obtained demonstrate i) marked Fpr2 expression in myeloid cells; ii) a lower yet sufficient degree of expression on stromal cells (fibroblasts and endothelial cells; iii) lack of modulation by disease or BMS235 treatment. These new data are presented in Figure EV1.

Q11. Is there a positive control - ? NSAID ASA or steroids

No. We did not include a positive control as our interest was on a potential direct effect of FPR2 in a newly established disease model. Also NSAIDs have been reported to be detrimental for RA patients with respect to diastolic dysfunction and HFpEF (reviewed in PMID: 33924323). In parallel projects we are testing anti-IL-6 and anti-IL-1beta approaches. At present, these experiments are ongoing yet we consider them to be out of scope of the present study.

Q12. Any adverse effects?

No adverse effects were observed. Of note, one of the markers for systemic arthritis is changes in body weight. As presented in Figure EV5, BMS235 did not alter body weight as compared to vehicle-treated mice.

Q13. Fpr2 null mice develop early onset diastolic dysfunction, which is associated with reduced survival upon ageing due to a prolonged failure of resolution responses (Tourki et al, 2020). This study indicates that Fpr2 exerts a tonic protective role against diastolic dysfunction. " Testing an activator of fpr2 is logical.

Thank you. We agree.

Minor comments

A. Figure 8 - the symbols upper left in the top left panel do not match the symbols in the figure.

We thank the Reviewer for this careful observation and corrected the figure accordingly. This figure is now Figure EV5.

B. The IL23R gene expression changes also look to be of interest -just a comment.

We fully agree with the Reviewer and will likely focus on this aspect in future studies.

C. Lung compliance measurement seems appropriate.

Flow cytometry is subjective albeit generally an accepted and an excellent tool - were the organs weighed.

Organs were weighed and measurements performed using standardized parameters for all experimental conditions.

D. Were histology samples taken for analysis of structural changes in heart and lung and arthritis?

Histology samples were not taken for further analysis of tissue. However, we have run a new experiment of HH+STIA, with or without BMS235 treatment, for the revision to generate samples for imaging.

We have quantified collagen fibres in hearts and lung using second harmonic generation with a two-photon microscope: the data are quite compelling and are presented in Figure 7. Treatment of mice with BMS235 significantly reduces presence of collagen fibres, with less conclusive results for the lung, likely due to the constitutive presence of collagen around the bronchioles and bronchi.

In addition, we have conducted spinning disk microscopy on 30- μ m thick sections of hearts and lungs from HH+STIA mice, monitoring immuno-reactivity for macrophages, neutrophils and fibroblasts. Whilst no significant changes in total cell numbers were observed with this

semi-quantitative method, we could spatially locate vimentin positive fibroblasts throughout the interstitium of the left ventricle. Neutrophils were sparsely seen and averaged around 4-5 cells per field. Galectin-3 staining was utilised to quantify pro-fibrotic macrophages which were elevated compared with naïve mice. Similar to the fibroblasts, macrophages were seen throughout the interstitial tissue. These analyses did not show a conclusive effect of the pharmacological treatment, probably because the experimental plan is underpowered, yet trends in cardiac cell numbers (Fig. 7G-I) were as expected.

In any case, we thank the Reviewer as this new set of experiments allowed visualization and distribution of cells pivotal to the experimental pathology, shown to be modulated by flow cytometry analyses (which obviously are more quantitative and allow deeper phenotyping through the use of a larger set of cellular markers).

Similarly to collagen fibre analyses, lung sections provided less clear-cut data: as an example, neutrophils could be found also in naïve lung sections. For completeness, we have still included the whole of these data (Figure 7J-L and Appendix Figure S6 and S7)

These new data are presented on page 11/12.

E. Define abbreviations such as E/A ratio on echo when first used
Done. See Page 5, second paragraph.

Referee #2 (Remarks for Author):

The manuscript by Margraf et al. provides an interesting approach for examining potential mechanisms for systemic complications of rheumatoid arthritis. This is a critical topic and one highly worthy of investigation. The manuscript is nicely written. To this reviewer, the approach has the added (potential) advantage of being able to examine complications in two organ systems. Is this potential advantage being acted up? This reviewer will highlight a few areas/suggestions and this will be followed by a series of questions clarification.

We thank the Reviewer for the positive comments to our newly developed model.

There are many strengths to this excellent manuscript. It is process/data of the selection of the Fpr2 target that requires clarification. In the results section the data in support of the target selection moves from human to rodent, from experimental to publicly available, all with differing degrees of "validation. Filling some of the gaps would be helpful. The section would benefit from highlighting/adding in "validations."

We thank the Reviewer for this feedback. Indeed, our goal was to incorporate both murine as well as human translational aspects.

Fpr2 was chosen as a target due to our previous study where Fpr2 agonism with the natural protein ligand modulated cardiac dysfunction in a genetic mouse model of inflammatory arthritis (Chen et al., 2021 in original and revised manuscript).

Q1. This reviewer is not confident in the logic that because Fpr2 KO mice have diastolic heart failure --- that agonist tone should "maintaining" normal diastolic function.

Understanding that there are more unknowns than knows, if there is "basal" activation and based on the data presented the receptor is up regulated, in the absence of loss of ligand then activity should be higher; not lower. The process of the selection is directly relevant regardless of the "outcome" of the target selection process; which appears to have been correct based on the therapeutic study. Detail on the target, its degree of activation, and cell specificity of expression/activation could go a long way to bridging that gap.

We have a long-dated interest in FPR2 as a master receptor that regulates several processes of the resolution phase of inflammation and have proposed that activation of this receptor may offer a novel therapeutic option to control pathologies with an inflammatory component, that means pretty much any pathology. FPR2 is highly expressed on myeloid cells but it is also readily detectable on other cell types including fibroblasts, endothelial cells and epithelial cells as well as smooth muscle cells. To corroborate this point, we have conducted new analyses on murine cardiac and lung cells, presented in Fig EV1.

We understand the reasoning of this Reviewer and the conceptual link between receptor expression and ligand expression, so that absence of ligand may lead to higher receptor expression. Similarly, if the receptor is expressed and functioning, and the ligand is there too, then it is unclear how we still have pathology. This classical association works well for several pharmacological pathways (e.g. neurotransmitter and neurotransmitter receptor) but it seems, to our experience, not to be applicable to resolution pharmacology.

We agree that a phenotype of the Fpr2 KO mouse indicates a tonic regulatory role for the ligand/receptor pair and therefore the ligand must be there: in fact, as an example, endogenous Fpr2 ligands like Annexin A1, Lipoxin A₄ and Resolvin D1 are detected in the circulation of rodents and humans (see for example our study in Dengue Fever [PMID: 35293862] and also the study in STEMI patients [PMID: 31345784]). A further element is the 'location' of the ligand, so that blood levels may be of little relevance to tissue-restricted effects on target cells that express the receptor.

In any case, our reasoning is the following. Pro-resolving pathways (like those centred on FPR2) are operative to avoid over-shooting of the host response and therefore of fundamental importance to guarantee **physiological homeostasis**. In **pathology** though,

they are clearly insufficient and overrun by pro-inflammatory signals. In terms of therapeutic approaches, so far, the approach has been to intervene and block specific pro-inflammatory mediators or antagonize specific pro-inflammatory receptors. With resolution pharmacology we proposed to activate further pro-resolving responses (see our original review in PMID: 26478210): this means that whereas the given pro-resolving pathway may be intact, it still can be harnessed to afford control of the pathology by further activation. In other words, it can be seen as *increasing the dose to enable* efficacy when moving from physiological settings (modest or moderate cell/tissue activation) to pathological settings (high cell/tissue activation). The fact that delivery of BMS235 is pharmacologically effective indicates that there is still 'capacity for FPR2' to be activated and exert its biological effects.

Apology for this long winding narrative, but it remains an important concept that may need to be elucidated further perhaps: in our view, this argument is complex only in appearance, yet it requires a major switch from current anti-inflammatory pharmacology to pro-resolving pharmacology.

To address the Reviewer comment, at least in part and in addition to the reasoning already detailed in the original submission, in the revised manuscript we cited a study which reports FPR2 expression in human heart, further substantiating target selection (Bouhadoun et al., 2023 in revised manuscript). In addition, we have run a new experiment of HH+STIA to quantify Fpr2 cell expression in hearts and lungs from HH+STIA mice, observing the expected presence on both immune and stromal cells. These new data are presented in Fig. EV1.

It is noteworthy that chronic administration of BMS235 over a two-week period **does not** alter/reduce FPR2 expression either on blood cells (Figure 4B) or heart/lung cells (new Figure EV1). The same outcome was also observed in our first study with KBxN F1 mice, where prolonged delivery of AnxA1 did not reduce nor alter FPR2 expression on circulating leukocytes or Fpr2 mRNA expression in the hearts (Chen et al., 2021).

Q2. The second area that requires attention is the homocysteine administration for heart failure with preserved ejection fraction model. There have been long-standing questions on the value/applicability of any mouse models in this area. The general consensus now would be that the model from Dr. Joseph Hill's lab (UTSW), has gained traction (for today) as the best model. Asking the authors to reproduce all of the data in this more established heart failure model is not reasonable. Perhaps selected parts could be repeated. The citation and another recent paper are:

Nitrosative stress drives heart failure with preserved ejection fraction - PubMed ([nih.gov](https://pubmed.ncbi.nlm.nih.gov/))
Mouse Model of Heart Failure With Preserved Ejection Fraction Driven by Hyperlipidemia and Enhanced Cardiac Low-Density Lipoprotein Receptor Expression | Journal of the American Heart Association (ahajournals.org).

The focus of our work is secondary organ injury in arthritis. As such, models of HFpEF per se (including the one of Schiattarella and Hill indicated by the Reviewer) are not that relevant. Similarly, one could have indicated models of myocarditis or diabetes-induced dilated cardiomyopathy: all these experimental protocols model different clinical scenario of human HFpEF. Our interest remains HFpEF in arthritis. Testing our target or tools in models without arthritis would be out of scope for this study.

However, we agree with the Reviewer on the need to test hypothesis in a different experimental model. Following a discussion with the Editor, we have performed new experiments and tested BMS235 and C43 on the dilated cardiomyopathy that develops in the KBxN F1 transgenic mouse colony (Chen et al., 2021 of the original manuscript). These new data are presented in Page 9 and Fig. EV2 and discussed in Page 16, end of first paragraph. Of interest, both compounds afforded cardio-protection in the absence of any effect on joint disease: we note that with aging KBxN F1 mice develop chronic arthritis,

evidently less responsive to treatment than HH+STIA animals. On the other hand, this new dataset in KBxN F1 mice corroborates the hypothesis that direct effects on cardiac cell (fibroblasts) and myeloid cell recruitment may underline the beneficial functional effects of Fpr agonists on the heart.

Minor areas for clarification:

1. It would be valuable know normal homocysteine levels and levels of the mice in this study, perhaps against human values. Information can be found that 10-12 micromol/L is the value for normal/control human subjects and 400 or greater for patients with homozygous abnormalities.

We have spotted a mistake and now present the correct values on Page 27, for both HH and HH+STIA. In addition, we now include values for KRN and KBxN F1. It can be appreciated that arthritic mice present higher values of circulating homocysteine, further recapitulating a feature of the human disease as presented in the narrative of the manuscript.

For completion, untreated C57Bl6 mice present circulating levels of homocysteine of $170 \pm 48 \mu\text{M}$ (N=6).

2. Data in the paper looks at the distribution of cells in heart/lung based on the use of serum injection with homocysteine in diet. The analysis by flow has the advantage of being quantitative but misses the opportunity to understand the spatial distribution of the cells, which could be useful. Also, this could also overcome the limitation on the success of accessing cells (i.e. macrophages) out of the tissue. Also, percents do not relate to total numbers which could be obtained with this "low tech" approach.

We thank the Reviewer for this critical feedback. Indeed, our primary intention was to study the quantitative regulative aspects of our disease model. Spatial approaches such as histology can again feature bias (selection of sections, staining protocols and more), thus we tried to choose a more objective approach. With regard to percentages vs. absolute numbers, we indeed tried to incorporate as many absolute values as possible (lung and heart), whereas this is more difficult in the joint as the whole tissue is not digested.

In any case, we have performed SHG and spinning disk microscopy in new tissue sections from HH+STIA mice and these data are shown in Figure 7 and Appendix Figures S6 and S7. Altogether, we observed quite a widespread distribution of fibroblasts in interstitium of the left ventricle, with macrophages present in the same areas.

Making strong conclusions from different analytical protocols is always risky, however we note that, for example, the larger number of neutrophils in lungs vs. hearts, as quantified by flow cytometry (10^5 versus 10^3 , respectively; see Appendix Fig. S3C vs. Figure 5A), is reflected by a larger number of neutrophils counted in lung sections (~200; Figure 7J) over heart sections (~4-5 neutrophils per field of view; not shown).

3. Studies were sex matched but were both sexes studied.
Yes, we have specified when male or male+female mice were used.

4. Supplemental table 2 - Adiponectin 650000 +/- 0? Correct.
Thank you. Corrected.

5. The sample size Stats on cytokines might need to be reviewed.
We are not sure what this query is about. If it is Supplementary Table 2, these are data from a multiplex plate analysis conducted on the samples generated from the animals as indicated at the bottom of the Table.

Referee #3

In this study, the authors established the murine model of rheumatoid arthritis complicated with both cardiac and pulmonary disfunctions by the combination of K/BxN serum transfer (STIA) and dietary supplementation of homocysteine (HH). The cardiac dysfunction induced in this STIA+HH model resembled HFpEF, frequently found in RA patients, histologically characterized by the infiltration of neutrophils and the expansion of specific fibroblast subsets, while the pulmonary dysfunction with reduced compliance accompanied by the increased M1/M2 macrophage ratio in the interstitial tissue. By analyzing public RNA-seq data from RA and HC PBMCs, the authors focused on FPR2, a GPCR for the pro-resolving mediators such as resolvin D1 as a therapeutic target. The authors examined a FPR2-selective agonist, BMS235, and a dual FPR1/2 agonist, C43. BMS235 ameliorated arthritis, and cardiac and pulmonary dysfunction, but C43 did not, except for the cardiac dysfunction. The authors demonstrated that C43 failed to improve lung compliance because of its FPR1-mediated potentiation of CCL2-mediated monocyte chemotaxis.

The manuscript is well-described. However, several points need to be clarified before the manuscript can be considered for publication.

#1. The authors previously found that K/BxN F1 mice developed HFpEF with the infiltration of activated T cells and the expansion of specific fibroblast subsets and that recombinant annexin A, an endogenous ligand of FPR2, ameliorated HFpEF. In this study, the authors established the murine model of RA complicated with HFpEF and reduced lung compliance. This inducible model by K/BxN serum transfer (STIA) and dietary supplementation of homocysteine (HH) may be helpful in investigating the specific gene function using KO mice with a B6 background. However, several concerns should be addressed: 1) HFpEF in the STIA+HH model was histologically characterized by the infiltration of neutrophils, which was different from the infiltration of activated T cells found in K/BxN F1 mice. Which HFpEF resembles that often found in RA patients?

Literature in RA is very scarce except immune cell characterization in the synovia, synovial tissue and blood. The number of mechanistic studies conducted in the heart and lung of RA patients, aside reporting functional alterations and higher incidence of HFpEF or idiopathic lung fibrosis, is minimal if not none.

Clearly, this remains a good point for discussion though we can only rely on the data we have. As the Referee correctly indicates, in KBxN F1 mice there is definitively a B cell component as well as a T cell component and indeed we have reported changes in these cell types (Chen et al., 2021 in the original manuscript). We are also aware that it is risky to make strong conclusions based on analyses conducted at specific time-points. As we have now tested BMS235 in KBxN F1 mice (new Figure EV2), it is tempting to suggest, if not conclude, that the amelioration achieved with the FPR2 agonist in the two models may indicate that neither T cells nor neutrophils are the sole primary cell target for therapeutic efficacy. This would leave macrophages and fibroblasts as main target cell candidates in the present experimental conditions.

In the revised manuscript, we have added new data generated with human MØ and synovial as well as cardiac fibroblasts and studied the effect of BMS235. These new experiments with human cells (new Figure EV4) add to the translational potential of the study, adding to the work conducted with human monocytes and presented in the original Figure 7 (now Figure 8).

2) The authors demonstrated reduced lung compliance with the increased M1/M2 macrophage ratio in the interstitium assessed by flow cytometry. In addition to these analyses, histological and/or imaging analysis by micro-CT should be performed to evaluate the interstitial pneumonia pattern found in the STIA+HH model.

We have conducted novel imaging experiments of freshly prepared samples of hearts and lungs from HH+STIA mice, treated or not with BMS235. These data are presented in Figure 7.

#2. The authors were interested in FPR2 by DEG analysis using public RNA-seq data from RA and HC PBMCs. However, FPR2 is a well-known receptor of pro-resolving mediators, and many studies, including the author's preceding study, demonstrated its anti-inflammatory and cardioprotective roles. Why did you focus on FPR2? Are there any rationales to select FPR2 from the unbiased DEG analysis?

We have a long-dated interest in FPR2 as a master receptor that regulates several processes of the resolution phase of inflammation and have proposed that activation of this receptor may offer a novel therapeutic option to control pathologies with an inflammatory component, that means pretty much any pathology. FPR2 is highly expressed on myeloid cells but it is also easily detectable on other cell types including fibroblasts, endothelial cells and epithelial cells as well as smooth muscle cells. Following the study with Annexin A1 in the KBxNF1 model (Chen et al., 2021), testing of the small molecule agonists was a sensible step for our team, as we are interested in determining the potential for resolution-based therapeutic tools. Moreover, comparing the selective FPR2 agonist with the dual FPR1/FPR2 agonist, offered a conceptual advance on the need for selectivity.

In other words, while there remain a vast number of potential targets, we selected a target which was feasible to be addressed by our experimental skills and aligned with the focus of our research.

All these efforts align with our aim to promote resolution pharmacology as a new branch of therapeutic innovation for clinical benefit of patients affected from RA.

#3. The reduced lung compliance in the STIA+HH model was alleviated by BMS235 but not C43. The authors found elevated numbers of Ly6ChighCCR2+ classical monocytes in the C43-treated lung and demonstrated that C43 potentiated the CCL2-mediated monocyte chemotaxis. This mechanism may explain why C43 had no effects on lung compliance even though both BMS235 and C43 had a similar effect on the M1/M2 macrophage ratio. However, the molecular mechanisms responsible for the effectiveness of BMS235 are more important and should be addressed in detail both in vivo and in vitro.

In our view we have provided sufficient in-vivo data to underpin the macroscopic beneficial effects of BMS235: reduction of neutrophil and fibroblast activation in the heart; reduction in pro-fibrotic macrophages and classical monocytes in the lung.

We have now extended the work with human cells, thus augmenting the translational value of the study, and studied the effect of BMS235 on macrophages, fibroblasts as well as macrophage/fibroblast co-cultures. The data obtained indicate that: i) BMS235 can hardly impact on human macrophage polarization in vitro; yet ii) BMS235 can regulate macrophage reactivity to achieve downstream modulation of selective markers of fibroblast activation. The latter point seems interesting and genuinely novel in relation to the properties of this molecule, and confirms a study with dermal fibroblasts and scleroderma [PMID: 31552041].

In addition, such cellular effects dovetail with the observations made in the murine heart and offer at least one mechanism that underpins the protection afforded against diastolic dysfunction in arthritis.

These new data are in Figure EV4 and discussed on page 13/14 and page 17/18.

#4. The authors demonstrated that BMS235 but not C43 attenuated arthritis. The data is inconsistent with previous data reported by Kao et al., 2014. The reasons for this discrepancy should be discussed.

Thank you, we discuss the potential reason for this discrepancy. While superficially the two studies present some similarity, there also present several differences including our

application of chronic settings (3-week arthritis) against the sub-acute settings of Kao et al (9 days). We also note different doses and routes of administration. Notably, at 6 mg/kg i.p. C43 was inactive in the study of Kao et al. 2014. We used 10 mg/kg p.o. here, which is probably equivalent in terms of exposure and plasma peak concentration.

On Page 17 we state:

“This is apparently different from Kao et al., who reported antiarthritic properties of C43 in settings of STIA (Kao et al., 2014): different time-points (3-week arthritis here versus 9-days arthritis), doses (10 mg/kg here versus 30 mg/kg) and route of administration (oral here versus i.p.) can account for the different effects.”

In the revised manuscript we present novel data with BMS235 in KBxN F1 mice showing protection against cardiac dysfunction in the absence of modulation of joint disease, corroborating the hypothesis that protection against secondary organ injury occurs at least in part through mechanisms dissociated from the degree of pathology of the arthritic joint. Thus, pathogenic mechanisms in secondary organs in settings of arthritis, or in arthritic patients, can be targeted therapeutically irrespective of the pharmacological control of joint disease. And this is one of the take-home messages of our study.

25th Feb 2025

Dear Prof. Perretti,

Thank you for submitting your revised study. We have now received the reports from referees #1 and #2 who evaluated your revised manuscript and your responses to all 3 initial referees. As you will see from the reports below, they are satisfied with the revisions, and I will therefore be able to accept your manuscript once the following editorial issues are addressed:

1/ Manuscript text:

- Remove the blue font and only keep in track changes mode any new modification.
- Please remove "not shown" (p. 13) as per journal policy, all results mentioned in the text must be shown in main or supplementary figures.
- Methods:
 - o Please update the section title to "Methods".
 - o Thank you for providing the reagents and tools table. Please remove the URLs below the table (all information should be contained in the table).
 - o Antibodies: please provide dilutions/concentrations for all experiments.
 - o Human samples: Include a statement confirming that informed consent was obtained from all subjects and that the experiments conformed to the principles set out in the WMA Declaration of Helsinki and the Department of Health and Human Services Belmont Report.
 - o Cells: please indicate whether the cells were tested for mycoplasma contamination.
 - o Please add a Graphics section, mentioning the use of Biorender: "(some of the... OR Figure #... OR synopsis) Graphics were created with BioRender.com".
- The Data Availability section should be placed directly after Methods, before Acknowledgements. Please only keep "Original files for flow cytometry, echocardiography are archived at <https://www.ebi.ac.uk/biostudies/studies/S-BSST1861>, whereas histology images are available at <https://www.ebi.ac.uk/biostudies/bioimages/studies/S-BIAD1635>." The remaining text should be removed.
- The funding information needs to be part of Acknowledgements. The text in the Comments box needs to be removed and these funders need to be provided as separate funders via More Funders option; to indicate 2 or more people, Control should be used for Mac users and Ctrl for Windows users
- Author contributions: CRediT has replaced the traditional author contributions section because it offers a systematic machine readable author contributions format that allows for more effective research assessment. Please remove the Authors Contributions from the manuscript and use the free text boxes beneath each contributing author's name in our system to add specific details on the author's contribution. More information is available in our guide to authors.

2/ Figures and Appendix:

- Please remove the legends from the individual figure Files (main and EV figures)
- Please note that we can now accommodate more than 5 EV figures, in case you would like to make some of your Appendix figures more accessible to the readers.
- Figure callouts: callouts for Appendix Figures and Table are not the same in all places; "Supplementary" needs to be removed and "Appendix Figure S#" and "Appendix Table S1" needs to be used throughout the text.
- Table EV1 needs to be removed from the manuscript and uploaded separately; EV Tables should roughly fit onto one A4 page. You could instead make this table a Dataset EV1, as an excel file (with the legend in a separate tab).
- The Appendix file should be in PDF format, with page numbers.
- If images/figures/figure panels were re-used, this needs to be stated in the figure legend (i.e. Fig. 5C and Appendix Fig. S1C; Fig. 7I and Appendix Fig S6; Figure 7L and Appendix Fig. S7).
- Please address the queries from our copy editors:
 1. Please provide the exact p values (in the figures or their legends) for figures 2B, C, E, F, G; 3A, D, E; 4B, D, E; 5A-E; 6A, C, F, G, H, I, J; 7A, B, E; 8B, EV2 A, B, C; EV3 A, EV4 B, EV5 C, D, E;
 2. Please note that information related to n is missing in the legend of figure 4A.

3/ Checklist:

- please fill in the subsection on mycoplasma contamination, in "Cell materials".
- please fill in the subsection on inclusion/exclusion criteria, in "Experimental study design and statistics".
- please fill in the section on informed consent and WMA Declaration of Helsinki, in "Ethics".

4/ Synopsis:

Thank you for providing a nice synopsis image. Please resize it to 550 px wide x 300-600 px high, and make sure the text remains legible. A cropped portion of this image will serve as thumbnail for the table of content on our webpage. Please also provide a synopsis text, that should include a short stand first (maximum of 300 characters, including space) as well as 2-5 one-sentences bullet points that summarizes the paper (maximum of 30 words / bullet point).

5/ As part of the EMBO Publications transparent editorial process initiative (see our Editorial at <http://embomolmed.embopress.org/content/2/9/329>), EMBO Molecular Medicine will publish online a Review Process File (RPF) to accompany accepted manuscripts.

This file will be published in conjunction with your paper and will include the anonymous referee reports, your point-by-point response and all pertinent correspondence relating to the manuscript. Let us know whether you agree with the publication of the RPF and as here, if you want to remove or not any figures from it prior to publication.

I look forward to receiving your revised manuscript.

With kind regards,

Lise Roth

***** Reviewer's comments *****

Referee #1 (Comments on Novelty/Model System for Author):

further extensive analyses on collagen were completed in addition to the original complex study. the findings are of potential great interest in understanding cardiovascular and pulmonary disease and RA.

Referee #1 (Remarks for Author):

Excellent responses to the reviews. The additional analysis are also of clear interest in understanding the mechanisms associated with cardio pulmonary disease in RA..

Referee #2 (Remarks for Author):

The authors were quite responsive to the comments from this Reviewer and Reviewer 3. No additional comments.

The authors addressed the remaining editorial issues.

17th Mar 2025

Dear Prof. Perretti,

Thank you for providing your revised files. I am pleased to inform you that your manuscript is accepted for publication and is now being sent to our publisher to be included in the next available issue of EMBO Molecular Medicine!

With kind regards,

Lise Roth
